

# Amundsen Sea Embayment accumulation variability measured with GNSS-IR

Andrew O. Hoffman[1], Michelle L. Maclennan[2], Jan Lenaerts[2], Kristine M. Larson[3,4], and
Knut Christianson[5]

[1]Lamont-Doherty Earth Observatory, Columbia University, Palisades, NY, USA
[2]National Center for Atmospheric Research, Boulder, CO, USA
[3]Institute of Geodesy and Geoinformation, University of Bonn, Bonn, Germany
[4]Aerospace Engineering Sciences, University of Colorado, Boulder, CO, USA
[5]Department of Earth and Space Sciences, University of Washington, Seattle, WA, USA

**Correspondence:** Andrew Hoffman (aoh2111@columbia.edu)

**Abstract.** In order to improve projections of the future ice-sheet surface mass balance and the interpretation of the isotopic signals of past accumulation preserved in ice cores, it is critical to understand the mechanisms that transport water vapor to the Antarctic continent. Global Navigation Satellite System (GNSS) receivers distributed across Antarctica to monitor ice velocity and solid Earth motion can be used to understand accumulation, ablation, and snow redistribution at the ice-sheet surface. Here,

we present a forward model for reflector height change between the GNSS antenna phase center and the snow surface and an inverse framework to determine accumulation rate and near-surface firn densification from the reflector height time series. We use this model to determine accumulation at the sites of three long-term on-ice GNSS receivers located in the Amundsen Sea Embayment (ASE) and at a network of GNSS receivers deployed in 2007-2008, 2008-2009, and 2009-2010 austral summers. From the GNSS-IR accumulation reconstructions, we find that extreme precipitation dominates total precipitation and

that extreme event frequency varies seasonally. We use our GNSS-IR accumulation reconstructions together with reanalysis products to characterize the atmospheric conditions that promote extreme snowfall in the ASE. The blocking pressure systems that promote extreme accumulation on Thwaites Glacier are facilitated by tropical teleconnections, specifically convection that promotes Rossby waves trains from the Western Pacific, Indian, and Atlantic Oceans to the Amundsen and Bellingshausen Seas.

## 1 Introduction

The mass balance of ice sheets and glaciers is defined as the difference between ice discharge and surface mass balance (the sum of surface melt, sublimation, and snow accumulation). Currently, the Antarctic ice sheet is losing mass at an accelerating pace due to changes in ice discharge primarily in West Antarctica driven by oceanic warming and submarine ice-shelf melt (Velicogna et al., 2020; Smith et al., 2020). These changes in ice discharge have partially been offset by an increase in surface

mass balance over much of the Antarctic Ice sheet since the early 19[th] century (Kaspari et al., 2004; Frezzotti et al., 2013; Medley and Thomas, 2019). This positive trend in the difference between surface accumulation and surface mass loss from



evaporation, sublimation, and surface runoff is estimated to have led to the mitigation of ∼10 mm of sea-level rise in the 20<sup>th</sup> century (Medley and Thomas, 2019). In West Antarctica, where ice loss due to submarine melt has increased the most in recent decades, snow accumulation has the potential to impact and mitigate ice-sheet committed retreat (Davison et al., 2023).

Reduced models using constant surface mass balance and submarine melt rate forcing have suggested that the retrograde slope of the bed topography beneath the ice sheet in this region may promote unstable retreat (Weertman, 1974; Schoof, 2012). More recently, Sergienko and Haseloff (2023) showed that the spatial gradients of accumulation, especially gradients near the grounding zone, and accumulation variability (in addition to submarine melt variability) impact marine outlet glacier response and the variability of grounding-line retreat. Accumulation gradients near the grounding zones of vulnerable outlet glaciers and

drivers of these gradients are thus important to measure and accurately represent in climate simulations used to characterize the response of Antarctic outlet glaciers to climate change.

Reanalysis remains the primary means of quantifying snowfall in Antarctica and has guided most of the atmospheric research and surface mass balance studies of West Antarctica in recent decades. Reanalysis techniques use archived observations with forecast models and data assimilation techniques to create gridded global datasets that describe the recent history of the

atmosphere, land surface, and ocean dynamics (i.e., Hersbach et al., 2020; Gelaro et al., 2017). The gridded nature of these products requires that they make coarse spatial assumptions for parameters such as vapor transport and snowfall that can vary spatially at scales that are relevant for glacier mass balance that are not well captured by reanalysis. Additionally, snow usually does not land where it falls in Antarctica (Lenaerts and van den Broeke, 2012). Reanalysis products typically assume that accumulation occurs in the location of precipitation. Even at the planetary scale, reanalysis products are not conservative. The

data assimilative techniques that are used to "nudge" solutions towards observations can introduce significant drift in moisture and energy budgets. Different reanalysis products also exhibit temperature and precipitation biases near the poles that vary between models. There are no long-term weather stations on Thwaites and Kohler Glaciers that record data assimilated into reanalyses, and there are very few studies that evaluate the performance of atmospheric reanalyses in comparison to observations. Ground-based observations of near-surface accumulation facilitate comparisons between reanalysis products that can

help distinguish physical processes from artifacts of these models and motivates model comparison with observations that are not assimilated directly into the reanalysis products.

Accumulation of snowfall, the primary positive contributor to surface mass balance in West Antarctica, has typically been measured using records of accumulation preserved in englacial stratigraphy in ice-penetrating radar data, ice-core layering, and surface elevation change recorded by sonic snow surface loggers and expressed in satellite altimetry (i.e., Kaspari et al., 2004;

Medley et al., 2014; Adusumilli et al., 2021). Regional changes in the observed past accumulation suggest that the processes driving the snow accumulation increase across the Antarctic ice sheet are more complicated than the simple temperature-snow accumulation relationships that follow from the capacity of a warmer atmosphere to hold more moisture (Fudge et al., 2016; Frieler et al., 2015). In addition to thermodynamic processes, snow accumulation variability has been connected to longer-term changes in atmospheric dynamics and circulation, and shorter-duration synoptic-scale dynamics that promote atmospheric

river landfall and extreme precipitation events (Dalaiden et al., 2020). Synoptic-scale moisture transport to West Antarctica in



particular represents the dominant driver of snow accumulation derived from atmospheric reanalysis in West Antarctica and accounts for almost 50% of the accumulation variability (Dalaiden et al., 2020).

Observations of discrete snow accumulation events in West Antarctica are important in order to further understand the synoptic drivers that cause extreme precipitation in this region, especially on Thwaites Glacier, where extreme events determined from reanalysis contribute over half of the total accumulation (Maclennan and Lenaerts, 2021). Extensive efforts have been made to obtain a number of continuous atmospheric observations of Antarctica since the International Geophysical Year (IGY) in 1957-1958, when 50 predominately coastal weather stations were first deployed across the continent. Following the launch of the first Collected Localisation Satellite in 1978 and the establishment of the Argos program that marked the start of data telemetry, several initiatives and programs began to erect permanent automatic weather stations (AWS) across Antarctica. Today, there are 267 AWS maintained across the continent. This still leaves much of the continent, and especially West Antarctica, sparsely sampled. Precipitation, a key variable for understanding surface mass balance, remains a notoriously difficult measurement to make and is still not routinely recorded across the AWS network. As a result, observations of accumulation in West Antarctica and Thwaites Glacier to date have also relied on records encoded in snow and firn, as preserved in firn and ice cores and englacial stratigraphy imaged by ice-penetrating radar.

## 1.1 Past Accumulation Studies in the ASE

The upper basin of the Thwaites Glacier was first visited for firn coring and near-surface radar data collection in the 2000-2001 austral summer as part of the International Trans-Antarctic Scientific Expedition (US-ITASE, Kaspari et al., 2004; Spikes et al., 2004). This ground-based traverse collected ice-penetrating radar profiles and a series of shallow firn/ice cores across the upper reaches of Thwaites and Pine Island Glaciers. Dated accumulation chronologies developed from the shallow core records and extended using englacial radiostratigraphy indicated that there had been an increase in accumulation ($\sim 5 - 10\%$) since 1970 in the western sector of the Pine Island–Thwaites drainage system and a slight decrease ($1 - 9\%$) in accumulation in sites near Marie Byrd Land and farther west (Kaspari et al., 2004; Spikes et al., 2004). The post-1970 increase in accumulation was strongly correlated with sea-level pressure, and was linked to the intensification of cyclonic activity in the Pine Island–Thwaites drainage system (Kaspari et al., 2004). These first accumulation rate histories in the ASE were limited primarily by the spatial extent of the radar data available to extend firn and ice-core accumulation chronologies, which are inherently point measurements, across the basin.

The collection of several high vertical resolution airborne radar datasets between 2009 and 2021 greatly expanded spatial coverage of radar data available to determine accumulation rates (Medley et al., 2013, 2014). Additional ice cores collected in the ASE also provided more highly resolved accumulation records in locations with higher accumulation rates. In contrast to previous results, neither observed nor modeled accumulation rates showed any trend since 1980 (Medley et al., 2013). Hypsometric distributions of accumulation rates revealed gradients in accumulation that change with elevation and distance from the Antarctic coastline. These gradients were found to be smaller than those simulated in reanalysis products, especially on Thwaites Glacier (Medley et al., 2014). Medley et al. (2013, 2014) did not attempt to determine the cause of the accumulation hypsometry, perhaps due to a relatively short (thirty-year) and poorly resolved (roughly decadal) record to compare with



atmospheric conditions in reanalysis products. They noted that further in-situ constraints from local, highly temporally resolved data may resolve discrepancies between modeled and observed accumulation rates (Medley et al., 2013). Although future advances in radar technology may enable retrieval of accumulation records with high temporal resolution deeper into the past, current highly-resolved radar-determined accumulation records in time and space are limited to roughly the past 40 years.

These methods rely on assumptions for snow deposition and densification and can convolve surface mass balance sig-
nals with integrated thinning and thickening signals associated with the settling and sintering of snow grains. Although these methods resolve annual and, more recently, seasonal changes in snowfall (Jones et al., 2023), they cannot determine relationships between accumulation events and atmospheric dynamics and moisture transport mechanisms that evolve on sub-seasonal timescales. Single snowfall events do not directly impact the long-term behavior of polar ice sheets because each event represents only a small fraction of the integrated decadal to millennial surface mass balance forcing required to engage the intrinsic
modes of marine outlet glacier response (Christian et al., 2020; Robel et al., 2018). However, they can be used to understand and link snowfall events to understand moisture transport mechanisms that change on longer timescales that are relevant for understanding the glacier's response to climate change and climate variability. This may be especially true in the ASE, where extreme precipitation events modeled in reanalysis account for more than half of the total precipitation (Maclennan and Lenaerts, 2021).

Recent work has suggested that extreme precipitation events may be responsible for substantial accumulation on coastal portions of Thwaites Glacier (Maclennan et al., 2022b). Prior observational studies have focused on determining mean annual accumulation rates from firn cores and radar stratigraphy (). These studies have also hypothesized linkages to regional atmospheric conditions and suggested teleconnections to tropical Pacific variability; however, the records examined lack the temporal resolution to actually interrogate the mechanism responsible for accumulation, determine whether accumulation varies
seasonally, and quantify the fraction of total precipitation associated with extreme precipitation events. Robust, high temporal resolution measurements of snow accumulation together with reanalysis data can elucidate relationships between accumulation and atmospheric conditions that could not be determined by past observations and linked modeling analyses.

## 1.2   GNSS Interferometric Reflectometry

GNSS interferometric reflectometry (GNSS-IR) offers an exciting method to determine accumulation rates with high temporal
resolution in remote and undersampled regions of the West Antarctic interior. We divide this manscript into two parts. First, we focus directly on the GNSS data used in this study (Figure 1). We first introduce the GNSS sites and the GNSS interferometric reflectometry method. Then, we describe the inverse method used to infer accumulation rates from the reflector height time series and discuss the seasonality and extreme events expressed in this new record. In the second part of this manuscript, we relate the GNSS-IR time series to synoptic-scale meteorological conditions in the southern Pacific and their connection to
tropical teleconnections. We first define and index extreme precipitation events observed in the GNSS-IR record. We then use the timing of these events to explore climate reanalyses to understand these highly localized records in the context of synoptic meteorology and planetary dynamics that promote precipitation in the region.





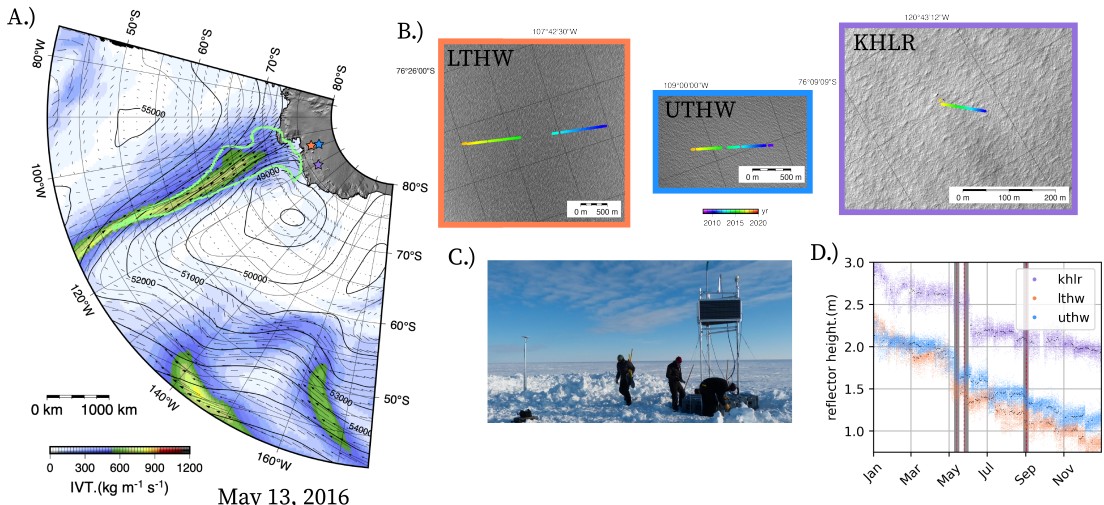

May 13, 2016

**Figure 1.** (A) Overview map of long-term GNSS-IR sites (LTHW in orange, KHLR in purple, and UTHW in blue), integrated vapor transport calculated between 300 hPa and 900 hPa pressure levels, the 500hPa geopotential height (contours), and an atmospheric river event detected on May 13, 2016. (B) Panels show the receiver position as a function of time. (C) Photo of the LTHW receiver and power system (credit: POLENET team). (D) Reflector height change in 2016 for all three long-term GNSS sites. Gray bars indicate the timing of atmospheric river events that made landfall in the ASE.

The Global Navigation Satellite System (GNSS) uses coded signals transmitted by GNSS satellite constellations at different L-band frequencies to determine three-dimensional position estimates of GNSS receiver antenna locations. The timing of
received GNSS signals provides range information important for establishing antenna positions but is distorted by electromagnetic wave interactions with the neutral atmosphere, the ionosphere, and reflections off nearby surfaces before the wave reaches the antenna (i.e., multipath reflections). The distorted travel times associated with multipath reflections have conventionally been treated as noise and informed early GNSS antenna gain pattern designs, which minimized multipath signals at high satellite elevation angles. Larson et al. (2009, 2015) established that multipath embedded in the signal-to-noise ratio (SNR)
of GNSS positioning data contains useful information about the reflection surface geometry (i.e., the height of the antenna above the surface) and the dielectric properties of the surface. Many subsequent studies have shown that the distance between the reflection interface and the antenna phase center can be calculated and monitored over time from the SNR recorded at the receiver antenna in order to understand local surface evolution.

Despite the proven effectiveness of GNSS-IR methods to measure surface height and reflector amplitude changes associated
with snow deposition (Shean et al., 2017; Larson et al., 2020; Pinat et al., 2021), only two studies (Larson et al., 2015; Siegfried et al., 2017) have used GNSS-IR to link surface height changes to climate and multi-year precipitation variability on the ice sheets. This is largely due to challenges maintaining multi-year on-ice GNSS receiver networks that have traditionally been deployed as part of short-duration arrays. Over the past fifteen years, The GNSS receivers located in the ASE have been





maintained to support several different projects that investigate solid Earth deformation, glacial isostatic adjustment, and ice
flow (Figure 1, 2). More than ten years of nearly continuous GNSS operation also offers a unique opportunity to explore
atmospheric dynamics that promote and control precipitation and accumulation in the ASE. We use GNSS-IR SNR data from
three long-term sites: lower Thwaites Glacier (LTHW), upper Thwaites Glacier (UTHW), and Kohler Glacier (KHLR), as well
as GNSS campaign arrays that were deployed across Thwaites Glacier as part of geophysical surveys in the austral summers
of 2007-2008, 2008-2009, and 2009-2010 to link extreme precipitation events and synoptic-scale atmospheric rivers observed
in reanalysis to highly localized accumulation records at individual GNSS sites (Figure 2).

## 2   Part 1: From GNSS reflector heights to accumulation time series

### 2.1   Methods: GNSS interferometric reflectometry reflector heights

The reflector height time series are calculated using the workflow described by Roesler and Larson (2018) based on the
earlier work of Larson et al. (2015). Daily solutions for receiver position were calculated kinematically relative to the sites
of nearby base stations (TF01 for the campaign and WAIS for campaign sites; BACK and HOWN for long-term sites) using
differential carrier-phase positioning estimates from the track software (Chen, 1998; King, 2004; Hoffman et al., 2020). Daily-
averaged position solutions for the base station sites were determined in GAMIT/GLOBK following standard GNSS processing
methods for glaciological applications (King, 2004). The reflector height was calculated from the signal-to-noise ratio (SNR)
data assuming a planar reflector surface geometry. After direct signal effects are removed the SNR data for a single satellite
and receiver can be modeled as

$$SNR(e) = A(e) \sin\left(\frac{4\pi H_R}{\lambda}\sin(e) + \phi\right), \tag{1}$$

where $A$ is the amplitude, which depends on the transmitted GNSS signal power, the satellite elevation angle above the
horizon, the surface dielectric constant and its roughness, $H_R$ is the reflector height, $\lambda$ is the wavelength of the transmitted
GNSS signal, and $\phi$ is a phase constant. This representation means a periodogram can be used to estimate $H_R$ from the
dominant frequency:

$$f = \frac{2H_R}{\lambda}. \tag{2}$$

To resolve this frequency the antenna must be at least two wavelengths above the surface ($0.4 - 0.5$ m, for GPS L1 and
L2 frequencies). The elevation and azimuth angles are calculated from the broadcast GPS ephemeris. The dominant SNR
frequency is determined using a Lomb Scargle periodogram. On average, this results in  70 estimates of SNR frequency every
day, which can be related to reflector height using equation 2.

We eliminate weak signals for each time increment that are $0.3$ m from the median reflector height for that time increment.
Longer sample intervals for calculating reflector heights provide a more robust result due to the larger number of samples in





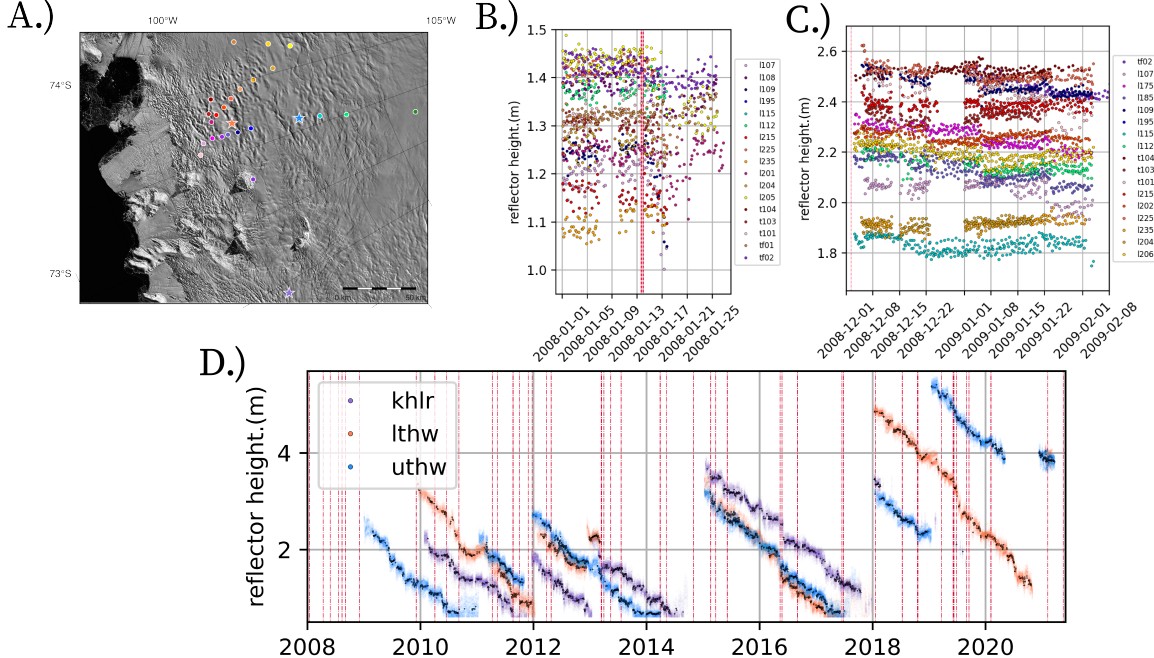

**Figure 2.** (A) Locations of all the GNSS sites (long-term sites indicated with stars and campaign sites indicated with circles) with reflector height observations from the (B) 2007-2008 array, (C) the 2008-2009 array, and (D) the long-term GNSS sites. For the long-term records, all reflection heights are shown with daily median reflectors in bold. Atmospheric river events are marked with red vertical dashed bands (see appendix section A2 for more details on event detection methods).

the increment. For the long-term sites, dense temporal sampling is not advantageous to spatially track snow accumulation due to sparse spatial sampling, so we use daily increments. The 3-hour increment was chosen to understand and estimate snow accumulation moving across the campaign GNSS array as storms moved up Thwaites Glacier. We then use the mean and standard deviation of the filtered signals (level 2 product shown in Figures 2, 3) to derive accumulation records (level 3 product
shown in Figure 4), which we describe in full in the next section.

### 2.1.1   Methods: Accumulation and densification forward model

To determine the accumulation record from observations of reflector height we use an inverse method (Metropolis-Hastings algorithm) that requires a forward model that translates the parameters of interest (in our case, an accumulation and densification time series) into observables (reflector height time series). Our forward reflector height model for on-ice GNSS represents
one-dimensional multi-layer snow compaction and grain settling from the surface to the depth of the GNSS antenna monument base. Each snow layer in the model is described by its thickness and density. Layers evolve following the compaction constitutive relation:



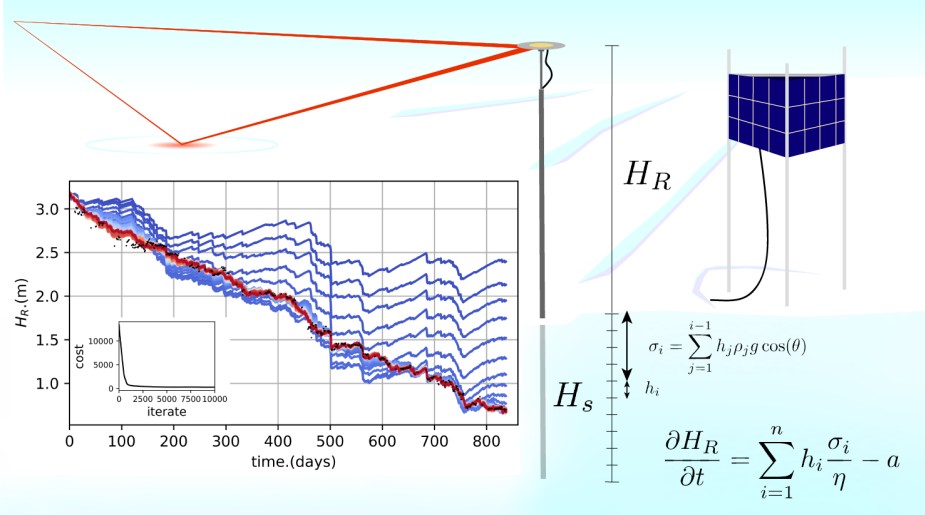

**Figure 3.** Model schematic along with the convergence pattern of the Metropolis-Hastings algorithm used to produce an accumulation time series.

$$\frac{\partial \epsilon_i}{\partial t} = -\frac{\sigma_i}{\eta}, \tag{3}$$

where $\epsilon_i$ is the layer strain, $\eta$ is the snow viscosity, and $\sigma_i$ is the overburden stress of the layers above. The vertical stress of the overlying layers is computed for each layer, $i$, as

$$\sigma_i = \sum_{1}^{i-1} g\rho_i h_i \cos(\theta), \tag{4}$$

where $\theta$ is the local surface slope, $g$ is the terrestrial gravitational constant, and $\rho_i$ and $h_i$ are the layer density and thickness, respectively. The snow viscosity, $\eta$, which describes how efficiently stress is transmitted through the column, has been linked to snow temperature, liquid water content, and snow grain geometry. Without in-situ observations of snow properties, we fix the viscosity for the duration of each reflector height experiment.

### 2.1.2    Methods: Reflector height inverse method

The number of unknown parameters in our model scales with the length of the reflector height record. We reset the inference of accumulation, viscosity, and initial snow density each time the antenna mast is raised (quasi-annually). We use Bayes' theorem to compute the distributions of accumulation at each time step and specify a likelihood model, assuming that observations of reflector height are independent in time and normally distributed around the true value at each data point with variance $\sigma_{R_{h_{obs_k}}}$. The objective functional defined by the least-squares difference between observed, $R_{h_{out_k}}$, and modeled, $R_{h_{obs_k}}$,

reflector heights for each increment, $k$, in the time series when observations are available is described by



The Cryosphere

Discussions

$$\phi_k(m_k) = (R_{h_{out_k}}(m_k) - R_{h_{obs_k}})^2, \tag{5}$$

which we use to define the likelihood of the modeled reflector heights, given the observations:

$$p(R_{h_{obs}}|m) \propto \prod_{k \in n} \exp\left(\frac{-B\phi(m_k)}{\sigma^2_{R_{h_{obs_k}}}}\right). \tag{6}$$

The priors for the accumulation record are modeled with Gaussian distributions with parameters listed in Supplementary Table S1. The Gaussian distributions for the snow accumulation are centered on an initial estimate of mean snow accumulation that takes the maximum of the difference of snow surface height divided by the mean snow density (300 kg/m$^3$) and a prior

of no snowfall ($300 \cdot \max(R_{h_{obs_{k+1}}} - R_{h_{obs_k}}, 0)$). With a likelihood model and a prior estimate of the model parameters, we use Monte Carlo methods to sample and thereby estimate the posterior distribution of the accumulation, density, and snow viscosity, and evaluate the statistical properties of the samples as a proxy for the joint posterior distribution of the model parameters. We use the Metropolis-Hastings algorithm to build estimates of the posterior distribution, map model degeneracy, and understand parameter uncertainty. The Metropolis-Hastings algorithm samples the prior estimate of each model parameter

and independently updates each parameter at each iteration by randomly drawing a sample from the proposed distribution. If the sampled posterior probability is greater for the proposed distribution than the current distribution, the new parameter value is accepted. Otherwise, it is accepted with probability proportional to the ratio of the current probability and that of the proposed value.

$$P_{\text{accept}} = \min\left(1, \frac{p(m_{i+1})}{p(m_i)}\right) \tag{7}$$

$$P_{\text{accept}} = \min\left(1, \frac{A\exp(-B\phi(m_{i+1}))}{A\exp(-B\phi(m_k))}\right) = \min\left(1, \exp\left(-B(\phi(m_{k+1}) - \phi(m_k))\right)\right) \tag{8}$$

Here $B$ is a tunable regularization parameter to facilitate convergence. We present the initial results of the time series below.

Then, in Part 2, we explore possible drivers of precipitation phenomena by connecting the events we observed to extreme precipitation in reanalysis.

### 2.1.3    Results: GNSS-IR accumulation time series

Using GNSS-IR, we create the first 10-year time series of snow accumulation at three sites in one of the most remote areas of West Antarctica: the Amundsen Sea Embayment. Currently, all precipitation on Thwaites at the sites of the GNSS receivers

falls as snowfall with no discernible rain events in the interior locations of the GNSS recorded in reanalysis. On Thwaites, precipitation is most frequent and intense in the austral winter (June, July, & August; JJA) with a peak accumulation in



August that tapers into the austral summer to a minimum in January (Figure 7). At LTHW, precipitation is ∼50% higher in austral winter than in austral summer. The seasonal signal is more muted at UTHW, with only a ∼30% difference between seasons. At the Kohler Glacier GNSS station (KHLR), peak accumulation occurs in March and is minimum in November.

Interannual variability and the seasonal cycle are driven primarily by changes in the frequency of extreme precipitation events. Following, (Maclennan and Lenaerts, 2021), we define extreme precipitation events using the 90th percentile threshold. Extreme events account for over half the total accumulation at all GNSS sites and explain 80% of the snow accumulation variability in reanalysis. These events also change seasonally (Figure 7). Over the observation time period (2009-2020), 13% of wintertime days (96 days), 6.2% of summertime days (45 days), 10.4% of springtime days (71 days), and 10.2% of fall days (82 days)

experience extreme precipitation.

Using the short-duration GNSS array with the two long-term Thwaites sites that were maintained for 1 month in the 2007-2008 austral summer and two months in the 2008-2009 austral summers, we also document the spatial patterns of snowfall across 20 stations in the ASE (Figure 2). During these shorter-duration campaigns, we observed 7 extreme precipitation events during the two summers (Figure 6). Across the short-term arrays in both summers, we find that it snows 2-4x more on lower

Thwaites (800 m elevation) than in the higher elevation sites (1200 m elevation) during extreme events (Figure 6). Gradients in accumulation are less strong on days when we do not observe extreme events. In the following section, we investigate the character of these extreme events and potential drivers with reanalysis.

## 3  Part 2: From extreme accumulation event detection to validation and event characterization in reanalysis

The remainder of this study is guided by two connected goals:

1. To test and further understand hypotheses that have been drawn from reanalysis alone without the aid of long-term GNSS-IR observations of snow accumulation. Specifically, we seek to verify recent studies that have found a strong seasonal cycle in annual snowfall in West Antarctica and studies that have identified extreme precipitation as a major fraction of total precipitation in the ASE (Maclennan and Lenaerts, 2021; Maclennan et al., 2022a).

2. To use our unique 10-year accumulation record to inform seasonal composites of pressure, surface temperature, and

wind anomalies preceding extreme precipitation events on Thwaites Glacier and evaluate the importance of blocking and tropical teleconnections on extreme precipitation in the ASE.

### 3.1  Data: Reanalysis

Due to the spatial and temporal limitations of in-situ observations available for the Southern Ocean and the WAIS, reanalysis products have been essential for examining spatial patterns and trends in atmospheric pressure, temperature, and moisture

transport in Antarctica. Reanalysis has helped identify the Amundsen Sea Low (ASL), a quasi-stationary climatological low-pressure system revealed in monthly average pressure anomalies located between the Antarctic Peninsula and the Ross Sea (Hosking et al., 2013). The ASL is one of three low-pressure systems distributed along the circumpolar trough (Turner et al.,





2013) and moves through the sector with the highest variability circulation in the Southern Hemisphere (Lachlan-Cope et al., 2001). The ASL also exhibits a pronounced seasonal cycle in location and depth (the geopotential height of the 1000hPa

pressure level Fogt et al., 2012; Hosking et al., 2013; Turner et al., 2013) that has been hypothesized to drive seasonal changes in moisture transport to the Antarctic interior. On average, the ASL depth is greatest during austral winter and weakest during the austral summer (Hosking et al., 2013). The center of the ASL also changes location seasonally, tending to be located farther northward and westward in austral summer and more southward and eastward in austral winter (Fogt et al., 2012; Hosking et al., 2013; Turner et al., 2013). The climate of West Antarctica and especially West Antarctic precipitation has been found to be

very sensitive to changes in the ASL (Raphael et al., 2016; Turner et al., 2013; Maclennan and Lenaerts, 2021). Because the low-frequency variability in ASL behavior has been connected to the low-frequency surface mass balance (Raphael et al., 2016; Turner et al., 2013; Hosking et al., 2013), understanding the drivers of multi-year variability and the dynamics of the ASL is important for understanding the committed response of rapidly changing outlet glaciers, such as Thwaites Glacier (Robel et al., 2018; Christian et al., 2020). This requires developing a deeper understanding of the synoptic and sub-synoptic scale cyclonic

systems that propagate through the Amundsen region and average together over time to form the low-pressure anomaly (Fogt et al., 2012).

Following many of the studies cited above, we use MERRA-2, RACMO2, and ERA5 reanalysis products to contextualize our observations of local surface height change from GNSS with large-scale circulation (Gelaro et al., 2017; van Wessem et al., 2018; Hersbach et al., 2020, Supplementary Figure S1). These products generally agree with one another; however, they

all underestimate the frequency of extreme events compared to the events recorded in the GNSS-IR records (Supplementary Figure S1). They are also poorly correlated with the GNSS-IR records (Supplementary Figure S1D, E, F). We chose to focus on ERA5 as simulated snow accumulation from ERA5 agreed with measured snow accumulation more than MERRA-2, and RACMO2 (Supplementary Figure S1).

ERA5 is the newest global atmospheric reanalysis product from the European Centre for Medium-range Weather Forecasts

(ECMWF) and has $0.25°$ latitude (28 km) by $0.25°$ longitude ($\sim 7\,km^2$) resolution (Hersbach et al., 2020). We focus our study on the Southern Pacific but include reanalysis data from ($20°$ N to $90°$ S) to evaluate tropical teleconnections that have been suggested to affect precipitation variability in accumulation records preserved in firn and ice cores.

### 3.2    Methods: Reanalysis composites

Using the GNSS-IR time series, extreme precipitation was classified using a seasonally corrected $90^{th}$ percentile threshold of

daily accumulation rates for lower Thwaites and upper Thwaites accumulation events. The union of these two indices was used to create a Thwaites Glacier catchment extreme accumulation event dataset. The results with Kohler included in this index do not change substantially; however, because of the observed offset in the accumulation seasonality, we exclude Kohler from the merged index of the sites on Thwaites Glacier. The GNSS-IR catalog of extreme accumulation events was used to calculate composite maps of geopotential height and wind anomalies at the 500 hPa pressure level, blocking anomalies calculated

from anomalies of geopotential height at the same pressure level, surface temperature anomalies (2 m), near-surface wind anomalies (10 m), integrated vapor moisture transport anomalies, and Rossby source anomalies using ERA5 daily averages.



Rossby source anomalies were calculated from 200 hPa winds following the methods described by (Trenberth et al., 1998) and (Holton, 2004), as implemented by (Dawson, 2016, Supplementary Figures S6, S7). Daily average outgoing longwave radiation (OLR) anomaly composites and sea surface temperature anomalies were calculated from NOAA SST data and OLR satellite data (Figure 11, Supplementary Figures S2, S4, and S5; Liebmann and Smith, 1996). Composites were calculated for the ten days preceding the onset of extreme precipitation and corrected for anomalies calculated relative to seasonal mean conditions. The interval of ten days was chosen to evaluate the importance of both proximal dynamics near the Amundsen Sea and tropical and extratropical teleconnections that precede changes in the ASE by days to weeks (Li et al., 2015, 2021). As part of this analysis, we also built catalogs of blocking events and the atmospheric rivers that made landfall in the ASE, both of which have been connected to extreme precipitation and changes in the surface elevation of the West Antarctic ice sheet (Adusumilli et al., 2021). More information on the methods used to create these catalogs can be found in the appendix (Appendix section A1).

### 3.3 Results: Composite investigation of reanalysis

### 3.3.1 Summer and winter time blocking in the Amundsen and Bellingshausen Sea

The mean blocking conditions from the days preceding the onset of observed extreme accumulation at the GNSS sites for summer (DJF) and winter (JJA) events are illustrated in Figure 8. We found that between two days prior to (day -2) and the onset of snowfall (day 0), blocking is present in the Bellinghausen and Amundsen Sea (defined by bounds 55°S, 75°S, 120°W, and 45°W) during 42% of winter extreme precipitation events and 37% of summer extreme precipitation events. Snowfall events with atmospheric blocking have statistically higher mean precipitation rates than those without blocking.

The blocking frequency anomalies associated with both summer and wintertime extreme precipitation events reveal significant blocking of the westerly flow in the Bellinghausen and Amundsen Seas before and during extreme snowfall events (Figure 8). The composites of 500 hPa winds are also consistent with blocking over large areas of the Bellinghausen Sea during extreme snowfall events. During the summer extreme precipitation events, blocking frequency anomalies are higher (note the difference in color scale) compared to wintertime blocking anomalies associated with extreme wintertime precipitation (Figure 8). Austral summer blocking anomalies also cover a significantly larger area than the austral winter anomalies and extend westward to much of the Amundsen Sea. In the austral winter, anomalously negative blocking frequency is observed closer to the Thwaites Glacier catchment. This low-pressure austral winter anomaly drives anonymous northerly mid-level winds that impinge on the coast of West Antarctica and appear to strengthen along the eastern flank of the anticyclone.

The mean blocking frequency (2009-2020, Figure 10) across all seasons in the Southern Ocean is largest in the Amundsen and Bellinghausen Seas. Blocking frequency in this region increases and shifts zonally from the Amundsen Sea in the summer to the Bellinghausen Sea in the winter. The amplitude of blocking frequency change also increases with a maximum in winter and a minimum in the preceding fall. The blocking frequency anomalies associated with extreme accumulation events over the period of observations are highest during summer and spring when the total frequency of blocking is low. This clear pattern in blocking frequency anomaly suggests that blocking of moist mid-latitude westerly flow north of the Amundsen





and Bellinghausen Seas in summer and spring drives significant summer and spring precipitation. In the next section, we use composites of near-surface temperature and integrated vapor moisture transport to explore these relationships further.

### 3.3.2   Summer and winter time vapor transport in the Amundsen Sea

The mean surface temperature and vapor transport conditions on the days preceding the onset of observed extreme precipitation events we observe with GNSS-IR are illustrated in Figure 9. Positive near-surface temperature anomalies are present across

the Amundsen Sea and the southern Pacific two days before extreme precipitation is observed on Thwaites Glacier. These temperatures steadily increase and peak on the day of the extreme event with positive temperature anomalies exceeding $1.5°$ C (Figure 9). During austral summer (DJF), these near-surface temperature and wind anomalies appear to come from the northwest. During the austral winter (JJA), anomalously high near-surface temperatures increase in the Amundsen Sea from the northeast (Figure 9A, D).

The mean integrated vapor transport (IVT) associated with these events also shows enhanced values in the Amundsen and Ross Seas. The circum-Antarctic jet that transports the majority of high latitude moisture in the Southern Hemisphere is diverted southward expressed as a negative easterly IVT anomaly where IVT is highest (Figure 9B, E, C, F). This negative moisture anomaly is coherent with a positive westerly and northerly IVT anomaly at higher latitudes that extends into the West Antarctic interior (Figure 9C, F). In austral summer, the positive IVT anomalies are more uniform and diverted to higher

latitudes by similarly uniform negative westward IVT anomalies. In austral winter, IVT anomalies integrated over the area of the entire Pacific are stronger than austral summer; however, the spatial structure of the anomalies is more heterogeneous and insignificant at $95\%$ confidence.

   Atmospheric river event frequency also varies seasonally with larger events making landfall in austral winter months. These events make up only a small fraction of the extreme events that appear to control precipitation on Thwaites. Similar to the

conclusions of other studies (i.e., Maclennan and Lenaerts, 2021), we find that atmospheric rivers account for  $8\%$ of the total accumulation measured at the sites of the long-term GNSS receivers over the 10 years of available observations. In total, we identify 19 atmospheric river events that make landfall within 5 days of an extreme accumulation event observed with GNSS-IR (90[th] percentile). Some of these events are not necessarily distinct (i.e. they occur within three days of other atmospheric rivers and could be considered part of the same vapor transport anomaly). The twelve distinct events are shown in Figure 5.

### 3.3.3   Summer and winter teleconnections

Composite anomalies of sea surface temperature and geopotential height at the 500 hPa pressure level were analyzed across the Southern Hemisphere to posit and test the significance of tropical teleconnections that may drive extreme winter and summertime precipitation in the Amundsen Sea. In the winter, we find significant positive tropical Atlantic, southeastern Atlantic, and southwestern Indian Ocean sea surface temperature anomalies preceding extreme snowfall on Thwaites Glacier

(Figure 11, Supplementary Figure S3). We also identify a clear train of alternating high and low-pressure anomalies consistent with Rossby waves that migrate eastward building up to extreme precipitation events (Figure 12, Supplementary Figures S3, S4, S5, S6, S7). In the austral winter, these Rossby wave trains, as expressed in 500 hPa pressure anomalies, extend from large





anomalies off the coast of Africa in the eastern Atlantic 4-5 days preceding the extreme precipitation event (Figure 12). These waves propagate across the Southern Ocean to the Amundsen and Bellingshausen Seas. In austral summer, the anomalies

associated with extreme precipitation events are stronger and more significant in the Bellingshausen Sea and appear to be driven by significant summer sea surface temperature anomalies in the central Pacific, in addition to significant anomalies in the Atlantic and western Indian oceans (Supplementary Figures S2, S3).

## 4    Discussion

The character and strength of the Amundsen Sea Low, specifically the magnitude of the low-pressure anomaly and the location

of the anomaly, have historically been the climatological indices most well-linked to accumulation variability in the Amundsen Sea Embayment and have been used to suggest connections between synoptic atmospheric variability and West Antarctic precipitation. These indices represent a mean anomaly integrating pressure variations over many days and many different synoptic conditions, which complicates the interpretation and attribution of signals that might promote extreme events. Chittella et al. (2022) showed that the second principal component of the geopotential height at 850 hPa (which is correlated to the 500hPa

geopotential height analyzed in our study) is highly correlated ($r^2 = 0.9$) with the longitudinal position of the Amundsen Sea Low and explains $43\%$ of the extreme precipitation events observed over this period. Our data, in conjunction with interpretations of regional reanalysis and other studies that use similar reanalysis products (Maclennan and Lenaerts, 2021; Chittella et al., 2022), suggest blocking highs in the Bellinghausen Sea drive extreme vapor transport into the WAIS interior (Figures 5, 8, 10). This pattern has previously been shown to be responsible for high snowfall events over Thwaites Glacier (Maclen-

nan and Lenaerts, 2021; Chittella et al., 2022), and these modeling studies agree with accumulation observations from GNSS interferometric reflectometry presented here. Though, in general, reanalysis underestimates extreme event frequency.

The GNSS-IR time series suggests that enhanced snowfall events observed in the ASE are associated with the blocking of the westerly polar jet stream and resulting perturbations in the composite integrated water vapor transport. Patterns in vapor transport anomalies associated with extreme precipitation on Thwaites Glacier suggest that the polar storm track is blocked and

that storms are re-routed poleward with distinct southward transport toward the Amundsen Sea. This is similar to the behavior observed in the Northern Hemisphere, where mid-latitude high-pressure systems over Europe and the Eastern Atlantic have been observed to divert storm tracks that then promote precipitation over Greenland (Pettersen et al., 2022).

Blocking in the Amundsen and Bellingshausen Seas exhibits a seasonal cycle that may explain the frequency of extreme precipitation, which controls over half of the total accumulation in the Amundsen Sea Embayment. Indeed, $40\%$ of extreme

precipitation events were accompanied by blocking in the Amundsen/Bellingshausen sectors, and blocking frequency anomalies associated with extreme precipitation become more significant in summer and spring when blocking is generally less frequent than in winter. Increases in winter extreme precipitation event frequency compared to summer appear to be the result of Rossby wave breaking from convection northwest of the Amundsen and Bellingshausen Seas in the Atlantic, Indian, and western Pacific Ocean basins. These Rossby waves form a low-pressure anomaly to the west of Thwaites Glacier and a high-

pressure anomaly to the east of Thwaites Glacier, funneling moisture from the tropics between the cyclone and anticyclone.



In the austral summer, extratropical Rossby wave propagation still appears to play a role in routing moisture to the Amundsen Sea; however, the anomaly associated with the blocking high-pressure system in the Bellingshausen and Amundsen Sea extends further westward than the austral winter high-pressure system. This is consistent with Rossby wave propagation from the central Pacific and the significant positive sea-surface temperature anomalies observed there preceding extreme summer 380 precipitation.

Previous numerical modeling experiments investigating tropical teleconnections to the Southern Hemisphere have suggested that Rossby wave propagation to the Amundsen Sea is driven primarily from the Indian and Atlantic basins northwest of the Amundsen Sea (Li et al., 2014; Simpkins et al., 2014; Li et al., 2015). Li et al. (2015) used idealized CAM4 simulations to show that anomalous convection events in the extratropical ocean basins in the southern hemisphere promote Rossby wave propaga-385 tion that sums linearly in the Southern Pacific Ocean. This complicates connecting the pressure anomalies and mechanisms that promote blocking in the Amundsen Sea to Rossby source mechanisms from specific basins. There are likely contributions from many different basins. This may explain the more complex pressure anomaly pattern observed in the summer when extreme precipitation is preceded by positive sea surface temperature anomalies in the central and western Pacific Ocean in addition to the Indian Ocean basin. The pressure anomalies in winter, when it snows the most in the Amundsen Sea, resemble Rossby wave 390 trajectories initiated from the subtropical Atlantic. Li et al. (2014) found the trajectory of the wave, the ray path, to be similar in all seasons, first propagating poleward from the subtropical South Atlantic until waves were reflected by weak vorticity gradients in the polar region. In the austral summer, Li et al. (2015) found that rays continue propagating poleward until they dissipate at high latitudes; in austral winter, however, the rays interact with a second reflecting surface created by anticyclonic shear associated with the poleward flank of the subtropical jet and the equatorial flank of the mid-latitude jet stream. This 395 reflecting surface is sufficiently strong in idealized model representations of Rossby wave propagation to steer the ray path of the waves to the Amundsen/Bellingshausen region. Significant sea-surface temperature anomalies observed in the Atlantic and western Indian Ocean preceding extreme precipitation are both consistent with the propagation paths of eastward propagating Rossby waves modeled in the Southern Ocean (Li et al., 2015).

The advection of moist air masses by blocking anticyclones leads to precipitation in the West Antarctic sector and represents 400 the dominant positive contribution to the Amundsen Sea Embayment surface mass balance. The anomalous warm 2 m air temperatures and water vapor transport along the coast of the continent also suggest that these pressure systems promote the transport of warm air and clouds that can insulate the continent when the snow is deposited. This has implications for the interpretation of ice cores in West Antarctica and the assimilation of ice-core data into paleoclimate reanalyses because the fractionation of isotopes is typically related to the local temperature of the precipitating cloud. The seasonality in accumulation 405 we observe in the GNSS-IR time series and reanalysis suggests that austral winter conditions and associated inferences of atmospheric temperature are sampled more than austral summer conditions in firn and ice cores. Our results also suggest that archives from West Antarctica for all seasons are likely biased toward the conditions associated with extreme precipitation.

The blocking pressure field that has been identified here and in previous studies as promoting extreme precipitation has also been connected to melting in West Antarctica and may increase the likelihood of rainfall instead of snowfall at the surface 410 as the sector warms (Niwano et al., 2021). At present, the processes contributing water to the near-surface cannot sustain a



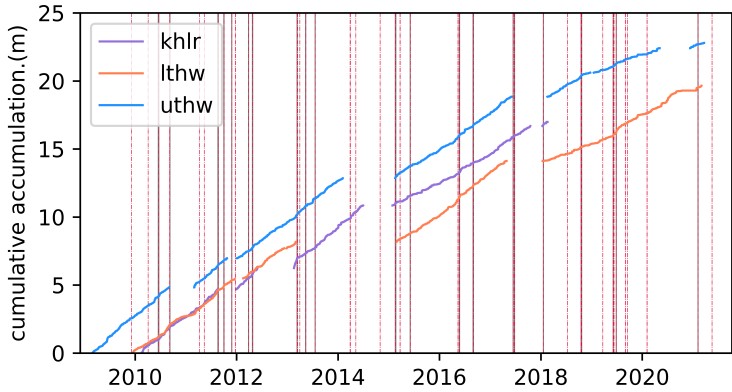

**Figure 4.** Observed cumulative accumulation at the LTHW, UTHW, and KHLR receiver sites with all atmospheric river events (dashed red vertical lines) and the events that occur within 2 days of the 90[th] percentile of daily accumulation rates (gray vertical lines).

perennial firn aquifer, and melt refreezes due to the cold content of the firn. Meltwater refreezing raises the temperature of firn, which can promote additional surface melt in successive events. Meltwater production and refreezing also deplete the firn air content, which has preceded hydrofracture-induced ice-shelf breakup on the Antarctic Peninsula. Changes in the firn air content associated with melting and strain-induced thinning have poorly constrained impacts on the future integrity and
susceptibility of the near-surface to hydrofracture and deserve further study. Another important implication of this work is the key role of processes that are poorly captured in General Circulation Models, which do not reproduce the recent increasing trend in geopotential heights in the Amundsen and Bellingshausen Seas and tend to underestimate extreme event frequency and blocking frequency in the broader southern Pacific region. This contributes to inaccuracies in the simulation of West Antarctic snowfall represented in General Circulation Models. These models can also have biases associated with meridional integrated
water vapor transport that underestimate snow accumulation in polar regions (Espinoza et al., 2018; Guan and Waliser, 2017). Our study suggests that in order to accurately model snowfall over West Antarctica and, in particular, the surface mass balance of Thwaites Glacier, atmospheric blocking, moisture transport, and precipitation processes must be properly simulated.

## 5   Conclusions

Spatially sparse in-situ meteorological observations of snowfall in Antarctica are an impediment to assessing the quality of
snow accumulation simulated in climate reanalysis products. GNSS-IR inverse methods provide a unique method to quantify surface mass balance and should be applied more broadly to GNSS networks in Greenland and Antarctica. These new techniques are advantageous compared to conventional automatic weather stations in several ways: (1) they require less power than AWS, (2) the observations represent a larger surface areas that is less likely to be biased by the station installation itself, and (3) these measurements can be made on GNSS sites that are deployed for other reasons, such as ice dynamic and solid



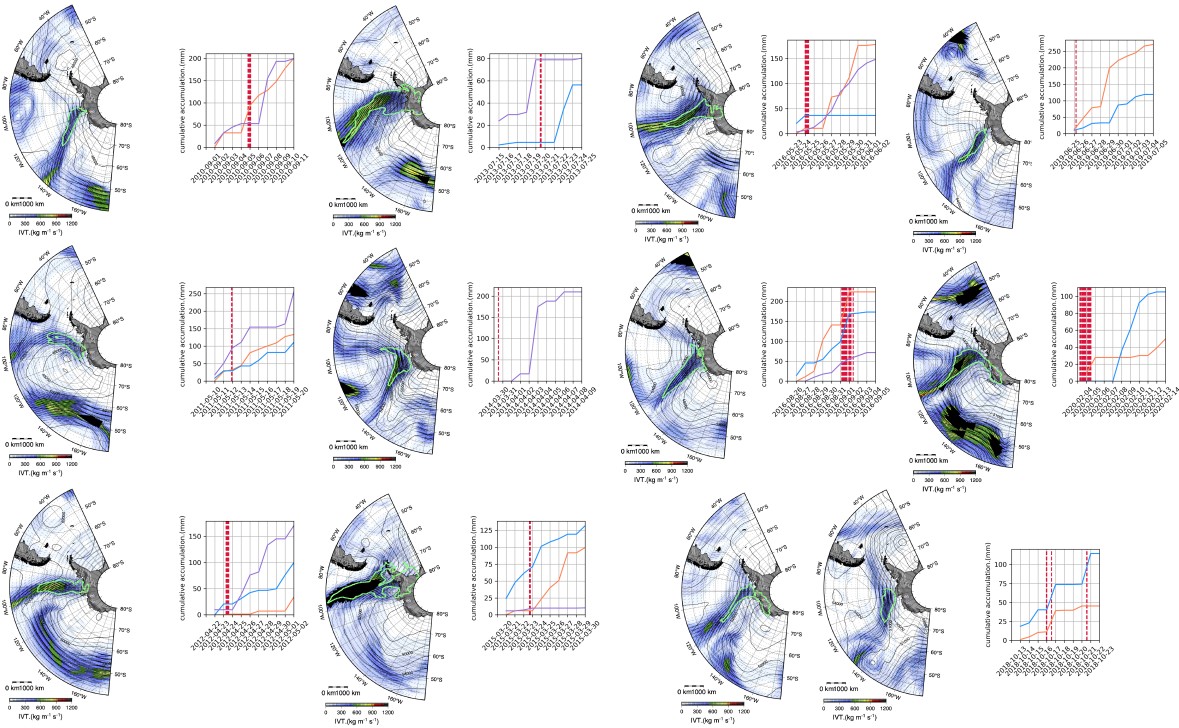

**Figure 5.** The detected atmospheric rivers with the guiding 500 hPa geopotential height and the integrated vapor transport for the events that occur within three days of the 95th percentile of height changes observed with GNSS-IR.

Earth studies and studies that have already been archived as part of historical geodetic surveys. In this study, we use GNSS-IR to derive accumulation records for the long-term on-ice GNSS sites located in the ASE along with three austral summers of campaign GNSS data. The GNSS-IR accumulation records reveal that precipitation on Thwaites is most frequent and intense in the austral winter (30 to 50% higher) with a peak accumulation in August that tapers into the austral summer to a minimum in January. The seasonal cycle is driven primarily by changes in the frequency of extreme precipitation events, which make up

over half of total precipitation in the Amundsen Sea.

We use the GNSS-IR snow accumulation time series to interrogate the synoptic atmospheric conditions that promote vapor transport to the Amundsen Sea that precedes extreme precipitation events. We find that blocking plays an important role in driving extreme precipitation. The connection between seasonal blocking in the southeastern Pacific and precipitation events in West Antarctica has important implications for the mass balance of vulnerable outlet glaciers located in the Amundsen Sea

Embayment. We find that accumulation in the Amundsen Sea varies seasonally and interannually and that much of this change can be explained by the presence of blocking high-pressure systems that promote extratropical cyclonic activity and vapor transport to the Amundsen Sea. This suggests that atmospheric blocking, moisture transport, and precipitation processes must be properly simulated in order to accurately model snowfall and surface mass balance over West Antarctica.


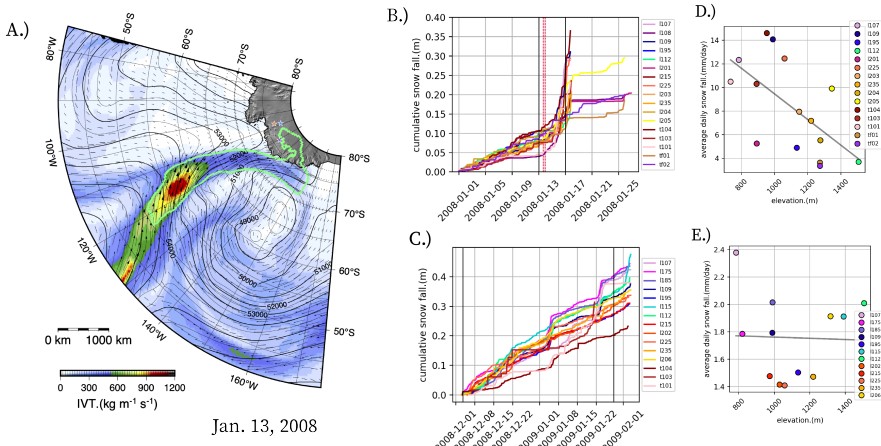

Jan. 13, 2008

**Figure 6.** (A) The guiding 500h Pa geopotential height (contours) and the integrated vapor transport for the 2007-2008 array atmospheric river event with the cumulative accumulation at the site of each receiver for the (B) 2007-2008 and (C) 2008-2009 seasons of array data. Average daily snowfall as a function of elevation for (D) the 2007-2008 and (E) the 2008-2009 austral summers.

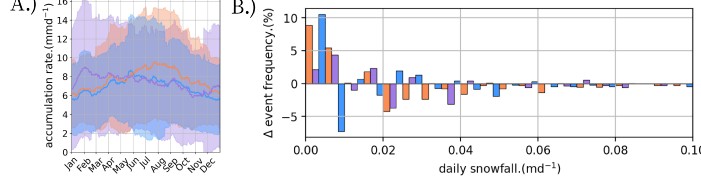

**Figure 7.** (A) Interannual accumulation and seasonal cycle across the lower Thwaites, upper Thwaites, and Kohler GNSS sites. (B) Distribution of the difference between seasonally high accumulation event frequency and low accumulation event frequency at upper and lower Thwaites (DJF-JJA) and Kohler Glacier (SON-MAM).

*Code and data availability.* Inverse methods are available at https://github.com/hoffmaao/thwaites-gnss-ir. GNSS reflection processing code
is available at https://github.com/kristinemlarson/gnssrefl.

*Video supplement.* A video of the accumulation event observed in the 2007-2008 summer (SupplementMovie.gif) is available in the Supplementary Materials.



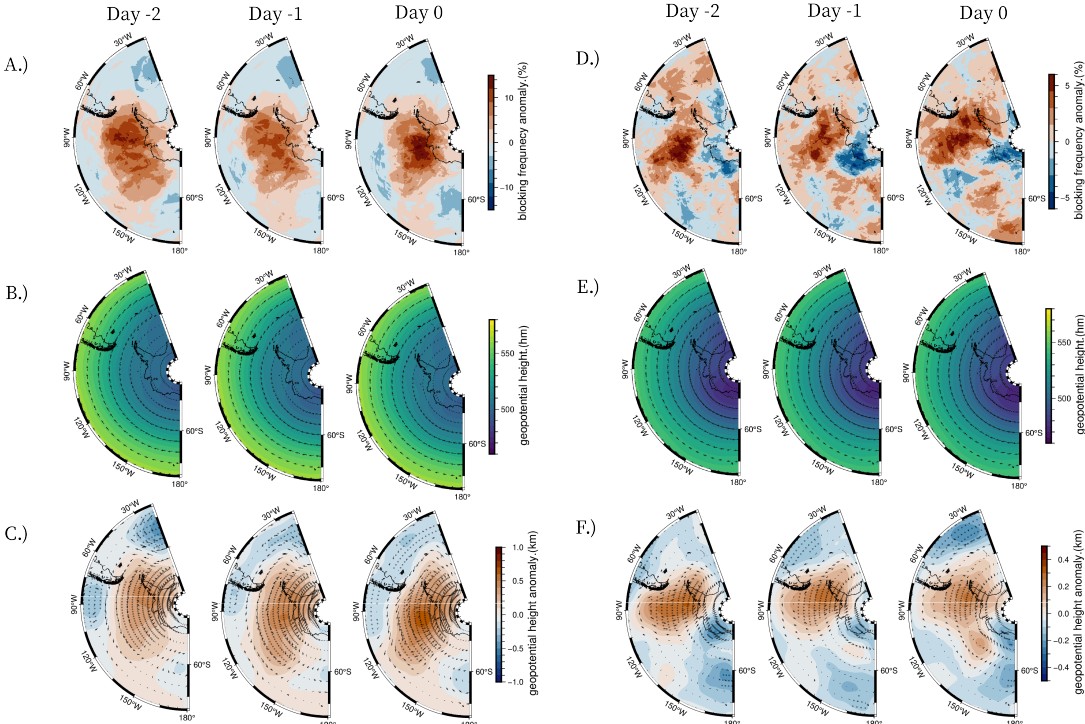

**Figure 8.** (A) blocking frequency anomaly, (B) 500 hPa geopotential height shown with vectors of the 500 hPa winds and (C) 500 hPa geopotential height anomaly and wind anomaly for events observed during the austral summer (DJF). (D) blocking frequency anomaly, (E) 500 hPa geopotential height and winds, and (F) 500 hPa geopotential height and wind anomaly observed during the austral winter (JJA). The day indices are relative to the day of precipitation onset at the GNSS receivers. Stippling of anomalies indicates significance at 95% confidence.

## Appendix A: Atmospheric catalogs

### A1    Blocking catalog

Atmospheric blocking is associated with stable shifts in the position of persistent high-pressure systems, which can impact the track of cyclonic storm systems. Blocking describes many different states of the atmosphere and thus many different methods have been developed to identify and track blocking events from reanalysis data (Schwierz et al., 2004; Barriopedro et al., 2010). We use the absolute geopotential height reversal-based (AGP) blocking index to identify blocking highs and track their evolution between 40°and 90°S, with a spatial overlap of a least five days based on the reversal of the geopotential height

gradient at 500 hPa. The index was first implemented by Steinfeld and Pfahl (2019) and is a two-dimensional extension of the classical blocking index developed by Tibaldi and Molteni (1990), which follows from the description of (Rex, 1950).





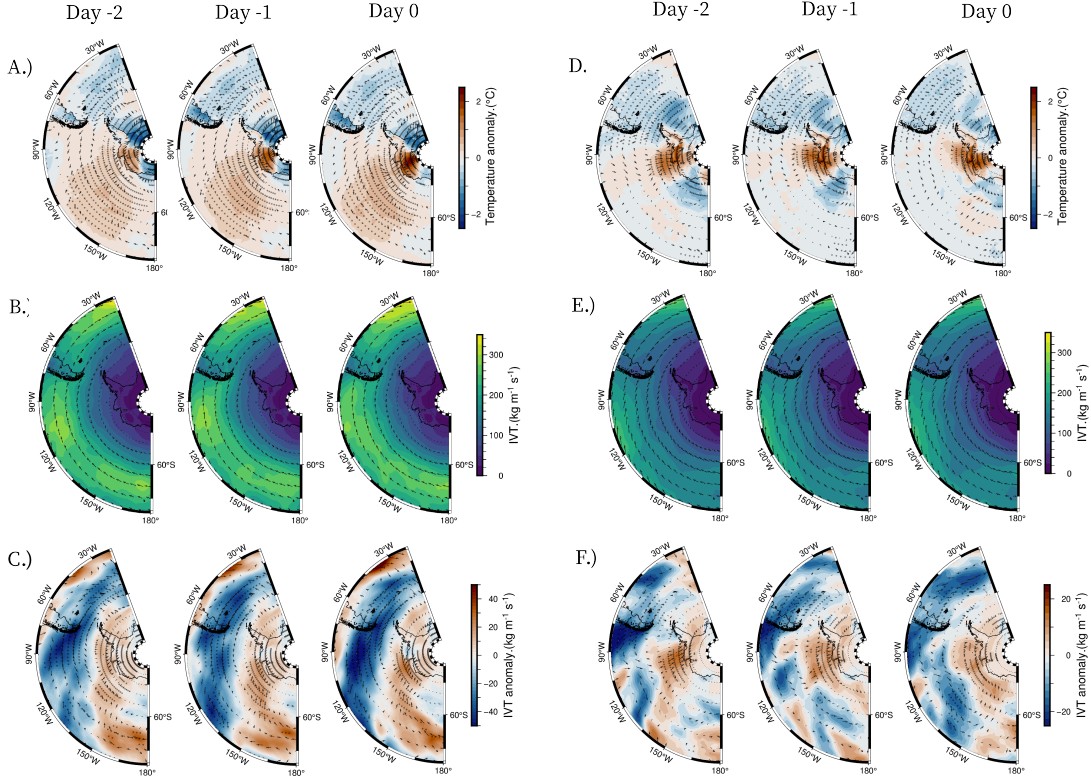

**Figure 9.** (A) near-surface 500hPa geopotential height for extreme summer (DJF) precipitation events on Thwaites, (B) geopotential height anomaly calculated relative to 30-day rolling mean, and (C) blocking frequency. The day indices are relative to the day of precipitation onset at the GNSS receivers. Stippling of anomalies in (A) indicates significance at 95% confidence.

## A2  Atmospheric river catalog

We also catalog the atmospheric rivers that made landfall in the ASE between 1979 and 2021. Atmospheric rivers were identified using vertical integrals of specific humidity and horizontal wind from ERA5 reanalysis to calculate total integrated vapor transport (IVT):

$$IVT = \frac{1}{g} \int_{surface}^{trop} qvdp + \frac{1}{g} \int_{surface}^{trop} qudp \tag{A1}$$

where $q$ is specific humidity, $u$ is the zonal horizontal wind component, $v$ is the meridional horizontal wind component, $p$ is pressure, and $g$ is the terrestrial gravitational constant. An event catalog was created using a threshold value defined monthly for each grid cell. This detection scheme first identified the 98[th] percentile of total IVT over a 5-month period centered on the month to be cataloged for each cell. The landfall location was then identified based on the intersection of the IVT anomaly with the Antarctic coastline. The axis of the IVT anomaly was calculated following Brands et al. (2017) and then used to



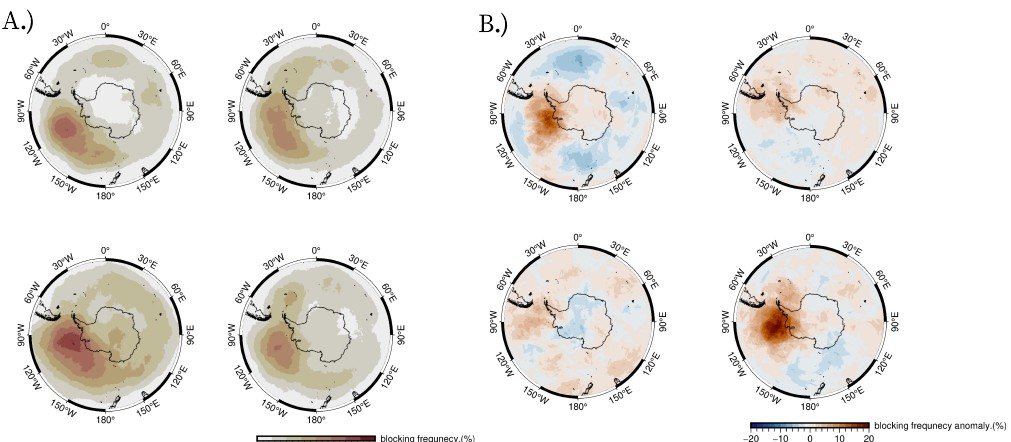

**Figure 10.** (A) Seasonal blocking frequency cycle and the (B) blocking frequency anomalies associated with extreme precipitation on Thwaites Glacier.

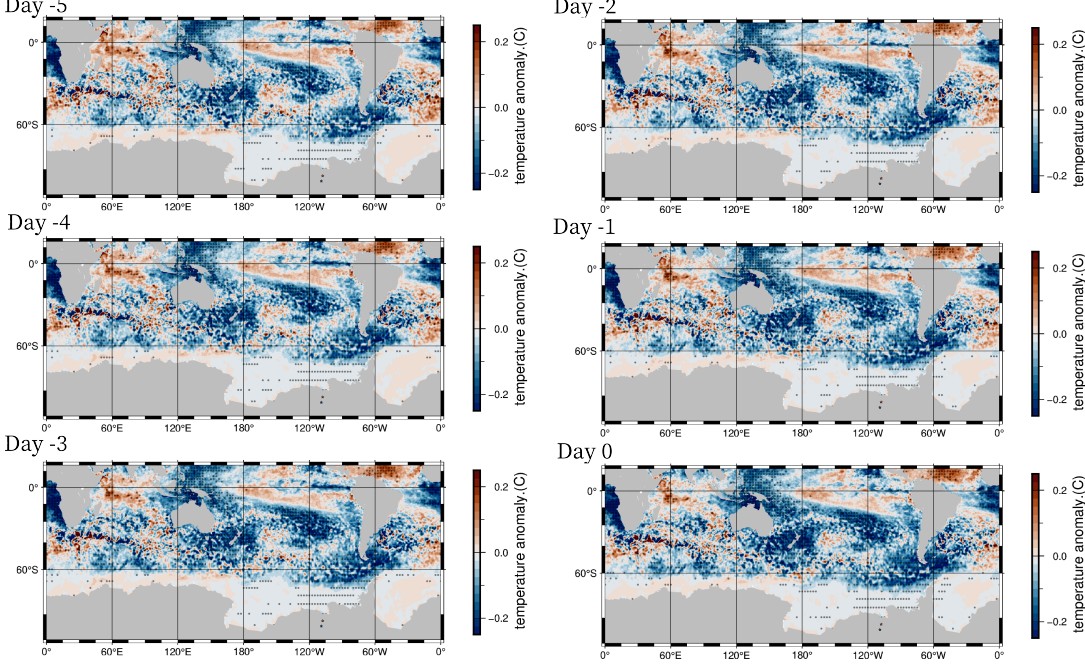

**Figure 11.** Austral winter sea surface temperature anomalies. Days for each panel indicate the shifted index for composite average prior to (or on) the day of extreme precipitation observed at the GNSS-IR stations located on Thwaites Glacier, which are plotted as orange and blue stars.

determine the length and aspect ratio of the anomaly along with the mean IVT direction and the mean IVT magnitude. Events were then filtered based on their geometry and the magnitude of the meridional component of IVT relative to a poleward

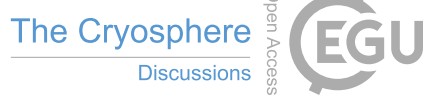

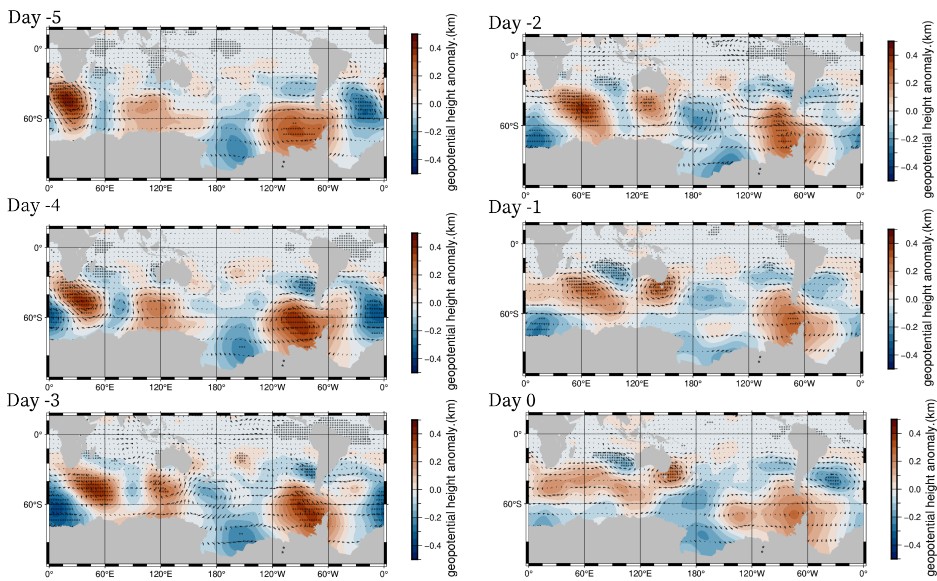

**Figure 12.** Austral winter geopotential height anomalies at the 500 hPa pressure level. Days for each panel indicate the shifted index for composite average prior to (or on) the day of extreme precipitation observed at the GNSS-IR stations located on Thwaites Glacier, which are plotted as orange and blue stars.

threshold value. Anomalies with a meridional to zonal aspect ratio greater than 2:1 and a meridional component of IVT larger than $100 kg \cdot m^{-1} s^{-1}$. The detection algorithm was applied to hourly fields of vertically integrated vapor transport from 1000

hPa to 300 hPa pressure levels to create an atmospheric river time series from 2008 to 2022 when continuous GNSS data on Thwaites Glacier were also available. Detected atmospheric river events presented here agree well with other catalogs made using similar methods (Wille et al., 2021).

*Author contributions.* AOH, MM, and JL theorized the study. AOH wrote the GNSS-IR accumulation inverse package, the atmospheric river event detection algorithm and implemented the blocking algorithm, MM derived accumulation histories from reanalysis for each of the

GNSS sites, KL wrote the GNSS-IR processing package. All authors contributed to the drafting and editing of the manuscript.

*Competing interests.* The authors declare no conflicts of interest.

*Acknowledgements.* We thank Terry Wilson, and the UNVACO/POLENET teams for maintaining the GPS sites, particularly Joe Pettit, Marianne Okal, Thomas Nylon, and Nico Bayou. The work was supported by a NASA FINESST award (grant 80NSSC20K1627), the NASA sea-level change team (grant 80NSSC17K0698), and the NSF-NERC International Thwaites Glacier Collaboration (grant OPP-1738934).





NOAA Interpolated Outgoing Longwave Radiation (OLR) data and NOAA Interpolated daily sea surface temperature (SST) data provided by the NOAA PSL, Boulder, Colorado, USA, from their website at https://psl.noaa.gov.



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
