# Peer review of "Amundsen Sea Embayment accumulation variability measured with GNSS-IR"

_The Cryosphere, 2023_

## Author Comment (AC3)

**General comments to editor and reviewers**

We appreciate the care that the reviewers took in responding to the manuscript. We especially want to recognize the efforts to collate relevant citations that the authors feel dramatically improved the manuscript.

We have also chosen, following the reviewer's suggestion, to build a more traditional structure for the paper. The study is no longer broken up into two parts, but five sections, including introductory material, methods and data, results, discussion, and conclusions with impacts and research context. We think providing a larger, more thorough introduction sets up the work that follows and aligns more strongly with the overarching goal of the study to show that GNSS-IR observations and methods we used to extract accumulation can be paired with interpretations of reanalysis that form the basis for the bulk of the discussion. The reviewer comments are bolded below. Our response is in standard font.

**Response to major comments provided by Reviewer 1**

**The notion that Amundsen Bellingshausen Sea blocking events drive marine air mass intrusions which results in major WA precipitation events has already been shown (Emanuelsson et al. 2018).**

We appreciate the reviewer mentioning this citation. The Emanualesson et al. ( 2018) study is directly related, and we now cite it in the introduction. Just to summarize for the others following the discussion, Emanuelsson et al. (2018) used a threshold pressure and duration to study anticyclonic activity on precipitation in the Ross sector. They compared annual precipitation simulated in reanalysis to ice-core records of annual accumulation preserved in the Roosevelt Island ice core. The Ross sector is part of West Antarctica, but precipitation (and in particular extreme precipitation) in the Ross and the Amundsen Sea Embayments are driven by different synoptic conditions (i.e., the conditions that promote extreme precipitation in the eastern part of the Ross sector do not promote extreme precipitation in the Amundsen sector). We have attached figures of the pressure system identified here and the pressure system identified by Emanuelsson et al. (2018) for reference. They are similar and they suggest that this dipole pressure pattern may be ubiquitous across Antarctica, but the results are themselves different, and the conclusions made from one set of observations require more complete treatment with modeled reanalysis to be extended to other catchments. To demonstrate these similarities, we applied our algorithm to accumulation measurements from a Ross weather station and replicate some of the observations from Emanuelsson et al. (2018).

[Figure]

[Figure]

In the first figure, we show the IVT anomaly associated with days leading up to extreme precipitation events (EPE) in the Ross. In the second figure, we show the pressure systems building to the day of an extreme precipitation event in the Ross. These figures show that the pressure systems that promote extreme accumulation in the Ross are

associated with blocking high-pressure systems in the Amundsen-Ross Sea at longitudes that are east of the blocking highs we detect in our analysis that promote extreme precipitation in the Amundsen sector. The fraction of extreme precipitation events at Thwaites Glacier is also very different from that at Roosevelt Island. See for instance, Turner et al. (2019). To observe the fraction of extreme precipitation events, we use daily resolved accumulation rates determined from our GNSS-IR observations in this study to composite EPEs on Thwaites Glacier. The ice-core data presented in Emanuelsson et al. (2018) is complementary in that it can be used to understand multi-year and decadal variability in precipitation, but the connection to atmospheric dynamics is less direct (i.e., the core cannot be used to composite extreme events as we did in this study with the GNSS-IR records).

We appreciate that the results of Emanuelsson et al. (2018) are relevant, and we have cited this paper in our study, but our data are used to think about atmospheric dynamics and processes happening over hours and days, not the integrated response of accumulation recorded in proxies analyzed by Emanuelsson et al. (2018). Our results have implications for records preserved in ice cores, as we have shown that the atmospheric composite associated with the majority of snowfall over thwaites is about two degrees warmer than the seasonal average temperature.

**You can tell the reader more clearly the mechanism that you suggest is at play. While it might be obvious to many readers of this journal, not everyone will know. The manuscript would become clearer with a hypothesis and the authors specify what part each reanalysis parameter that they use can tell us.**

This is a two component study that is building confidence in a new method. We are using an existing network that was deployed to understand horizontal ice motion to estimate snow accumulation. The goals are thus twofold. First, we seek to prove this method for event resolving accumulation time series. The second goal is analyzing these data to understand the drivers of extreme precipitation in the Amundsen Sea Embayment (ASE). We show that blocking conditions that appear to be connected to Rossby wave trains in the Indian and Atlantic oceans drive the majority of extreme precipitation in the ASE.

**The type of composite of plots non-anomalies that are presented in Figs. 8b, e and 9b, e are not meaningful. As the blocking events will not show an exact spatial overlap, the composite average will not show the anti-cyclone, reversal of the westerlies, characteristic of blocking that would appear if you looked at individual blocking events. Better than to show some examples of certain events if you want to show this type of figure. Like you do in Figure 5. You can discuss Figure 5 more, it is only referenced one time in the text. And then maybe remove the 8b,e and 9b,e panels. Increase the size of Figure 5 so that it is easier to see. Dedicate one page for this figure, and show it in landscape layout.**

The composite of blocking anomalies presented in Figures 8b,e and 9b,e are similar to those used in other studies, for instance, Pettersen et al. (2022), to understand the significance of blocking during extreme precipitation events observed over Greenland. We believe they are useful to include as a comparison to plots in that study, which may already be familiar to readers. They are not intended to reveal the pattern of wind anomalies, which we plot with the composite of geopotential height anomalies separately.

The geopotential height field fields (not the anomalies) were included to show the reference field used to make the anomalies in the wind and 500hPa geopotential height shown in the lower panel. They also capture/reveal seasonal changes in the geopotential height field that are relevant for interpreting seasonal anomalies. These can be moved to the supplement.

**The wind vectors in the figures are hard to see and it is hard to differentiate them and the stippling if you don't zoom in. If the stippling shows where the Z500 is significant perhaps you can show where the winds are significant by just displaying significant winds or by showing significant winds in a different color. And don't use black color for both the winds and the stippling.**

[Figure]

We agree with this comment and have made the stippling and the arrows different colors. Significant winds can be characterized by direction and magnitude. We focus on the temperature in these plots and include the winds to give context to the circulation with these near surface winds.

**107. Accumulation has been reconstructed in many studies using deep cores. Not just from firn cores, e.g., (Thomas et al. 2015; Winstrup et al. 2017). Emanuelsson et al. were able to illuminate the importance of blocking for West Antarctic (WA) airmass intrusions using annual dD and accumulation records from the RICE ice core. They investigated these relationships further using high-resolution ERA**

**records of precipitation and accumulation from AWG measurements from AWS (Emanuelsson 2016; Emanuelsson et al. 2018).**

We appreciate this suggestion and recognize that the ice-core community papers on accumulation reconstructions for West Antarctica were under-cited. We also note that there are significant differences from ice-core studies cited here and the kind of analysis that is possible with GNSS-IR records. The ice-core proxy records used in Emanuelsson et al. (2018) were not event-resolving records (i.e., these records do not record accumulation that can be attributed to individual pressure systems). They represent integrated signals on annual timescales or longer and cannot reveal information when there is not snowfall. The AWG measurements and GNSS-IR methods that we develop in our paper can be used for this purpose and show that blocking and extreme precipitation are strongly related. In a previous response, we have shown how these AWG measurements reveal composites leading to EPE for a AWG station located in the Ross Embayment.

The integrated signals recorded in ice cores may well reflect blocking in the Ross. However, this is not a hypothesis that can be tested directly using data that record accumulation anomalies on interannual timescales. In the introduction, we try to recognize the value that these long-term records hold. We've cited both of these papers here. We also think it's valuable to recognize some of the shortcomings of these records where GNSS-IR and AWG stations can potentially provide valuable insights. These data can resolve events and thus can interrogate extreme precipitation directly.

**213. Are these seasonal differences significant? How do they compare to a precipitation record obtained from ERA5 over the overlapping period and using the whole available ERA5 period (40+ years)?**

These seasonal differences are significant. These seasonal changes in precipitation have been well-studied using ERA5. See, for instance, McClennen and Lenaerts. (2021). This is the first direct observation of seasonal variations in accumulation in this region. Our observations suggest that the phase of seasonal changes in precipitation may depend on longitude. This is a new observation and has not previously been discussed or interpreted in the context of reanalysis.

**3.1 Data: reanalysis - There is plenty of material about ASL, but no introduction about blocking in the high southern latitudes. Plenty has been written about this and blockings linkage to the tropical Pacific, e.g. (Renwick and Revell 1999; Renwick 2005).**

We have included three sentences in the introduction on high-latitude blocks connecting literature on the Amundesen Sea Low (ASL) that we cited previously to the study mentioned in this review by Emanuelsson et al. (2018).

**L 238-256. This paragraph about the ASL is more suited for the introduction than under the header describing the reanalysis data.**

We have moved this paragraph to the introduction. We hope that by returning to a more traditional paper format to describe the method and then demonstrate and compare the method with results from reanalysis we make the flow and logic of the paper clearer.

**Specify how many percent of the whole data series are gaps. The KHLR time series seems to only be 8 years.**

KHLR site records data for 2941 days, the LTHW site records accumulation for 2941 days, and the UTHW site records accumulation for 3282 days. This has been added to the main text as a reference to a supplementary table that summarizes these accumulation observations with all available accumulation records in the low elevation regions of the Amundsen Sea Embayment.

**As there are so few events, 19, it would be reasonable to look at individual events and check their origin. As you do in Figure 5 but including latitudes farther north. Then you can check the origin of the wave trains without the risk that the pressure anomalies get cancelled in certain regions in the composite. That is if there is a high pressure in one region for one event and a low in the same region for another event they will get cancelled in the composite.**

There are 19 atmospheric river events. Several of these events span many hours/days and this is why we show only 12 map view figures. The same pressure systems often contribute significant IVT anomalies that qualify as high-latitude atmospheric rivers. There are many more extreme precipitation events than atmospheric river events. We have tried to clarify that extreme precipitation events are not the same as atmospheric river events.

**Bear in mind that you are just looking at 10 years. Considering decadal-scale variability, it could be that wave trains during the period could have a certain origin say mainly Atlantic, while if you had a record from the preceding 10 years there could be another main origin. Just due to decadal-scale variability.**

This is a great point, and we appreciate that the source variability may change. We do not have observations that resolve single events that extend back as far as the reanalysis records or ice core records. We could use hourly ERA-5 reanalysis to look at conditions that promote extreme precipitation further back in time. We choose to focus on the time period where we have GNSS-IR observations as these observations remain the focus of our study. We believe these records are complementary (and under used) and can bridge a gap between climate and atmospheric dynamics studies. We have shown that both the Pacific and Atlantic basins contribute to the variability we observe in

the ASE, and we have made it clearer that the conclusions on basin-specific tropical teleconnections are limited to the study period analyzed.

**259 to 263. These sentences are confusing and not precise. Be quantitative. Use statistics to back up your argument, p- and r-values. It is hard to see in the suppl. figure which reanalysis dataset is best. Can there be a benefit of showing the time series of the data, and comparing the measured data with the reanalysis data? And mark the extreme events. Like Figure 4 but with the reanalysis added. For example, by extracting time series of precipitation data from the grid point(s) that are closest to the sites. Something similar to fig. 1.11 in this PhD thesis (Emanuelsson 2016).**

We appreciate this suggestion. This was done in Mclennan et al. (2023). We have included in the supplement the relationship between reanalysis and GNSS-IR accumulation histories but increased the size of these plots to make the obscurity of this relationship more clear. The r values were not included in the supplement initially but have been added now.

**Is it hourly data that you use?**

We use hourly reanalysis. The GNSS-IR data are processed to increments that the user can specify. In this study, we use 3-hourly and daily increments to understand the short array records and the long term GNSS sites, respectively.

**Response to Minor comments provided by reviewer 1**

**Turner et al. have also highlighted the importance of large precipitation events (Turner et al. 2019).**

We have now cited Turner et al. (2019). Thanks for bringing this to our attention.

**Pg.1 L20. For accumulation increase seen in ice cores, you can cite Thomas et al. (Thomas et al. 2015).**

We cite the Thomas et al. (2015) later in the paper, but we've included it here as well.

**58. Delete "in order"**

Thanks for the suggestion.

**72. Cite Mayewski here too (Mayewski et al. 2005).**

We have included this citation. Thanks for the suggestion.

**74. …. from shallow ice cores and extended… does this sound better?**

We agree with the reviewer here. Thanks for the suggestion. We've changed this to. "Accumulation chronologies dated from the shallow ice cores and extended.."

**92. .. resolution that reaches further back into the past,…**

We've elected to delete this entire sentence. Rather than speculate about whether future technology can or will allow the determination of accumulation from layers at sub-annual resolution, we choose to focus on the method we developed that can quantify accumulation in this region using GNSS-IR.

**107. The references are missing.**

Thanks for noting this. We have deleted this section and elected to add a section that transitions into the outstanding questions and study overview.

**115. Thwaits is considered a coastal not an interior site.**

We appreciate this reviewer comment. This is true, and we recognize the potential confusion that can be caused by using Thwaites Glacier (which encompasses a large region from the grounding zone to the interior central ice-sheet divide). When we discuss interior Thwaites sites, we refer to the GNSS sites located hundreds of kilometers into the interior of the Thwaites basin, which are closer to WAIS divide (clearly an interior site) than to the grounding zone.

**116. Use the acronym.**

We thank the reviewer for the suggestion. We have made this change.

**119. In the Southern Ocean Pacific region.**

GNSS-IR can also be used outside of the southern Pacific with dual-frequency GNSS datasets across the globe.

**137. … due to challenges in maintaining…. GNSS receiver networks. GNSS networks have traditionally been...**

The second clause in this sentence has been removed.

**140. (Figures 1, 2)**

We have added the Figure 1 identifier.

**212. Present the figures in order of appearance. And close to where they are mentioned in the text.**

We appreciate the reminder. We have gone through the text and the figures are now mentioned in the order that they appear. We have also moved two of the figures that appear later in the text (near the acknowledgements/references) and moved these into the main text with stronger TeX formatting conditions for the figure positions.

**221. Split this up into two sentences and put this part into parenthesis "(with the two long-term Thwaites sites)"?**

We deleted the long-term Thwaites sites referenced within this sentence.

**228. I don't see the benefit in splitting up the method and results sections in this way.**

We were trying to demonstrate a method (as there are some readers who will be most interested in this aspect of the study) and prove it in an application specific to snow science/ accumulation and reanalysis. We agree though with the reviewer that this non-traditional framing made the paper harder to read and have reverted to a more traditional framing (introduction, data & methods, results, discussion and conclusion).

**L 237. Change the header to, Reanalysis data.**

See change. Thanks for the suggestion.

**L 242. Cite Rapheal's zonal wave number three paper (Raphael 2004).**

We've included the citation of Rapheal (2004). Thanks for the suggestion.

**L 243. … highest variability in atmospheric circulation in the Southern Hemisphere (Connolley 1997; Lachlan-Cope et al. 2001).**

We have included this suggestion.

**266. Suggestion: ...S). The northern limit is set this far north to be able to evaluate the possibility of tropical teleconnections…**

We've changed this sentence to:

We focus our study on the Southern Pacific but include reanalysis data from tropical latitudes ($20^{\degree}$ N to $90^{\degree}$ S) to evaluate the possibility of tropical teleconnections that have been suggested to affect precipitation variability in accumulation records preserved in firn and ice cores.

Thank you for the suggestion.

**276. Define IVT here at its first mention.**

We have added the integrated vapor moisture transport acronym here.

**292. 55°–75°S, 120°–45°W. Show this region as a box in a figure. Isn't 45W too far east to be considered the Amundsen Sea?**

This is the area defined as both the Amundsen and Bellingshausen seas. We have pluralized seas to make this more clear.

"..blocking is present in the Bellinghausen and Amundsen Seas.."

**333. So, if there are multiple atmospheric river events for one extreme precipitation event, do you disregard the event? That seems strange. Such an event would still indicate that rivers are important right? Is the last subplot in Fig. 5 with two composite maps from such an event?**

We do not disregard the event. These events are often long duration; some events last several hours to several days. The events are identified at hourly resolution, but were manually inspected, and where they were associated with the same synoptic pressure system, these events were counted as one event (i.e., there are 24 events for atmospheric rivers lasting a day detected using our algorithm). Extreme precipitation events are resolved at daily increments.

**337. Composite?**

The notion that these are composited events is signified by the second part of the sentence where we say that the anomalies precede extreme precipitation events.

**350. Emanuelsson et al. showed the importance of blocking for major and extreme WA precipitation events. And highlighted that these blocking events occur in an area with an average climatological low-pressure anomaly, the ASL (Emanuelsson et al. 2018).**

The Amundsen Sea Low moves substantially on seasonal timescales. The pressure anomaly that drives extreme precipitation on Roosevelt Island does not drive extreme precipitation in the ASE. Neither site is representative of extreme precipitation across West Antarctica. We are working on establishing the ubiquity of the dipole pressure pattern (high-pressure blocking system to the West and low-pressure system to the East) across Antarctica for different Antarctic catchments.

**354. EOF2, is this the PSA1, Pacific South American patterns (Kidson 1988; Karoly 1989; Mo and Higgins 1998)?**

This is the second EOF of pressure during extreme events. This was identified by Chitella et al. (2022), and does appear to resemble Kidson (1988), Karoly (1989) and Mo and Higgins (1998) (shifted to the west). We have now reviewed these other older more foundational papers. When we think the feature in these studies is the same feature identified in our study, we have included citations in the main text.

**405. Are the seasonality results significant? Considering that the record is only 10 years long and there are only 12 (19) events? If so this would be valid for Thwaits cores but not WA cores in general which can have other seasonal biases or no bias (Küttel et al. 2012). There can also be a temperature bias associated with these extreme events (Sime et al. 2009; Emanuelsson 2016).**

The seasonality results are significant. We also include the temperature recorded during extreme precipitation events to show that bias might induce affect temperature signal in the snow deposited at the surface.

**L 415-418. Do you have a paper that you can cite for the finding that blocking is not well-represented in models?**

Most CMIP5 models do not reproduce the observed blocking frequency in the North Atlantic sector (Vial and Osborn, 2012; Masato et al., 2013; Anstey et al., 2013; Davini and D'Andrea, 2016) with up to a 30–50% underestimation of wintertime blocking frequencies (Woollings et al., 2018). Many of the new generation models (CMIP6) show an improvement in reproducing blocking frequencies, but for some regions, such as the North Atlantic, most still have too little blocking (Davini and D'Andrea, 2020; Schiemann et al., 2020). These results also extend to Antarctica. See figure below from Liu et al. (2022) using CMIP6.

[Figure]

This figure shows differences of DJF blocking days at Z500 in GFDL models from observations (shading of at least 8 days) with a contour interval of 2 days (the zero

contour is omitted). The x markers represent the centers of blocking days in observations (black) and each model (red). CMIP6 models have the number 4 in the names.

**L480. Reference the NOASS datasets together with the other datasets in section 3.1. Or thank some more reanalysis providers here.**

We have included a recognition statement in the acknowledgments and also cited it here.

**Fig. 1. Do the red lines indicate the same as the gray? …., 2016 (light blue contour).**

The gray lines indicate extreme precipitation events detected with the GNSS-IR method (data), the red lines indicate atmospheric river events that were detected with reanalysis (model product). This has been added to the Figure 1 text.

***Fig. 5. Can you explain a bit better what we see in the figure, the contour indicates atmospheric rivers.***

The green contour indicates the atmospheric river. The colormap shows the Integrated vapor moisture transport that was used to identify it.

**Some of the text in the figures is very small. I don't think you need to have both a dot and parenthesis for the subplots in the figures.**

We have increased the size of the figure plots. Thanks for the suggestion.

**Figure 7. Explain the colors again. What is the shading for, std? What is the difference between light and dark orange? One is std and the other is the mean of the std for the two Thwaits sites? Split up 7a into three subplots as the shading from one site can hide the shading for the neighboring sites? Or is it enough to make the plot larger?**

The bars are the standard deviation for the ~10 year time series. The dark line is the mean. We've separated this into three plots so that the reader can more easily see the differences between each plot.

**Figure 12. The stars for the sites are hard to see here. Increase their size.**

 We have increased the size of the stars in the figure. Thanks for the suggestion.

[Figure]

**Caption Fig. s1. …" against accumulation determined from reanalysis products for (D) LTHW, (E) UTHW, and (F) KHLR GNSS sites."**

We appreciate the gentle suggested change and have made the edits.In general, we have also edited this paper to make it more legible. Thank you!

[Figure]

For the accumulation vs. reanalysis accumulation plots. How do you see that the ERA5 record is best? Provide some more statistics, r-values? Add a line in the plots and text boxes with r and p values. Do you compare hourly data from the different datasets? Perhaps you need to average the data over a longer period to make them comparable. Again plotting the measurements together with the reanalysis data would be good as a first step to confirm if they capture the same events and if the timing agrees.

We have included an r-statistic in the text. We agree that this is important. We have also added a new table in the supplement that captures the correlation between these products.

446. The supplementary animation looks interesting. Please provide information on what it is showing. Do you compare the GNSS measurements with ERA5 precipitation data over the period when you have good spatial coverage? This period with an expanded network seems unique and something that you could evaluate and discuss more.

The caption accompanying this animation is stored on the linked website where the view is hosted, but we have included it here as well. We do not compare these data directly to the reanalysis, but the agreement in event timing is strong across the array and establishes the utility of reusing/rexamining GNSS data that have previously been published (The array data were previously published by Fudge et al. (2015)).

**Response to comments provided by Reviewer 2:**

**check the numeration of all the figures: figures 6 and 8 are discussed before figure 5.**

We appreciate the gentle suggestion. We've gone through and made sure that all figure captions and labels come in the order that the figures are introduced.

**anticipate entering the parameter "B" of Equation 6, because it is introduced only many lines later (after Equation 8)**

The parameter B in equation 6 has been introduced closer in the text to the equation.

**Figure 1: the legend of panel B) (time scale) is too small, and the color legend doesn't seem coherent with the plotted receiver positions. In Panel D) could you explain the meaning of the dashed red lines?**

We have changed the legend of panel B to make this larger. The color is coherent with the receiver positions. In panel D, the red dashed lines are the timing of atmospheric river events, which are included in the text.

[Figure]

**Figure 2: what about the gray vertical lines?**

We appreciate this correction/clarification. This was not clear previously. The gray vertical lines are the timing of extreme precipitation events identified from GNSS-IR observations. We have added text that reflects this in the figure caption.

**Line 189: The authors indicate the variance with the Greek letter sigma, but usually square sigma is used. Also the quantities in equation 5) are called "least squares differences" but do you just mean "squared differences"?**

We can write this in terms of the standard deviation as you suggested. We also have changed to the squared difference.

**Figure 3 is not clear. In the figure caption, could you explain the quantities represented in the plot?**

We have changed the caption in Figure 3 to the following:

The model schematic and convergence pattern of the Metropolis-Hastings algorithm used to produce an accumulation time series. In the main plot, we show the convergence of the modeled RH (colored lines) and the observed reflector height time series (black points) as a function of model iterate. The objective function we use to score solutions is shown as an inset plot. The misfit of the model compared to observations decreases and converges as a function of iteration in the inverse procedure.

**Figure 5: in the xy-plots it's impossible to read the labels because they are too small. Furthermore, do the color lines in the plots refer to the LTHW, UTHW and KHLR GNSS stations? In this case a station legend should be repeated also in this figure. Do the vertical red lines represent the height changes observed with GNSS-IR? Why the thickness of the red vertical line is different?**

The color of the plots mirrors the colors that are used throughout the figures for each GNSS station. We refer to these labels in the figure caption. We also have made the entire plot larger (it now takes up a whole page), and we have reoriented the figure to work well within a vertically oriented page.

**Line 223: The authors affirm that "during the shorter-duration campaigns we observe 7 extreme precipitation events during the two summers (Figure 6)", but in Figure 6 these events are not represented.**

The extreme precipitation events are highlighted in gray now. The atmospheric river events are included in red.

**Figure S1: the diamond symbol that should represent ERA5 is not plotted in the panels. It seem to be substituted by "X" symbols. Moreover, do you refer to panel (D), (E) and (F) instead of (A), (B) and (C) in the sentence "Accumulation measured with GNSS-IR plotted against accumulation determined from reanalysis products for (A) LTHW, (B) UTHW, and(C) KHLR GNSS sites", aren't you?**

We appreciate the suggestion. The other reviewer also made this point about S1. The S1 figure has been made larger (see above). The diamonds did appear as x's in the supplementary figure plot, we've made the ratio of the width of these bars to the size of the diamonds larger to make them appear more distinct. We also have changed the figure caption so that it properly describes panels D, E, and F rather than A, B and C.

**Line 216: "Following, (Maclennan and Lenaerts, 2021)," —> "Following (Maclennan and Lenaerts, 2021)", without commas.**

Thank you for noting this mistake. We have changed this text to: Following Maclennan and Lenaerts (2021), ..

**Line 224: With "2-4x" do you mean "2-4 times"?**

Thank you for noting this. We have changed to 2-4 times.

**References**

*Anstey, J. A., Davini, P., Gray, L. J., Woollings, T. J., Butchart, N., Cagnazzo, C., Christiansen, B., Hardiman, S. C., Ospresy, S.M., & Yang, S. (2013). Multi-model analysis of Northern Hemisphere winter blocking: Model biases and the role of resolution. Journal of Geophysical Research: Atmospheres, 118(10), 3956-3971.*

*Connolley WM (1997) Variability in annual mean circulation in southern high latitudes. Clim Dyn 13:745–756. https://doi.org/10.1007/s003820050195.*

*Davini, P., & D'Andrea, F. (2016). Northern Hemisphere atmospheric blocking representation in global climate models: twenty years of improvements?. Journal of Climate, 29(24), 8823-8840.*

*Davini, P., & d'Andrea, F. (2020). From CMIP3 to CMIP6: Northern Hemisphere atmospheric blocking simulation in present and future climate. Journal of Climate, 33(23), 10021-10038.*

*Emanuelsson BD, Bertler NAN, Neff PD, et al (2018) The role of Amundsen–Bellingshausen Sea anticyclonic circulation in forcing marine air intrusions*

into West Antarctica. Clim Dyn 51:3579–3596. https://doi.org/10.1007/s00382-018-4097-3

Emanuelsson D (2016) High-Resolution Water Stable Isotope Ice-Core Record: Roosevelt Island, Antarctica: a thesis submitted to the Victoria University of Wellington in fulfilment of the requirements for the degree of Doctor of Philosophy (Geology) / by B. Daniel Emanuelsson. Thesis (Ph.D.)--Victoria University of Wellington, 2016.

Karoly DJ (1989) Southern Hemisphere Circulation Features Associated with El Nino-Southern Ocscillation Events. J. Clim. 2:1239–1252

Kidson JW (1988) Interannual Variations in the Southern Hemisphere Circulation. J. Clim. 1:939–953

Küttel M, Steig EJ, Ding Q, et al (2012) Seasonal climate information preserved in West Antarctic ice core water isotopes: relationships to temperature, large-scale circulation, and sea ice. Clim Dyn 39:1841–1857. https://doi.org/10.1007/s00382-012-1460-7

Lachlan-Cope TA, Connolley WM, Turner J (2001) The role of the non-axisymmetric antarctic orography in forcing the observed pattern of variability of the Antarctic climate. Geophys Res Lett 28:4111–4114. https://doi.org/10.1029/2001GL013465.

Liu, P., Reed, K. A., Garner, S. T., Zhao, M., & Zhu, Y. (2022). Blocking Simulations in GFDL GCMs for CMIP5 and CMIP6. Journal of Climate, 35(15), 5053-5070.

Maclennan, M. L., & Lenaerts, J. T. (2021). Large‐scale atmospheric drivers of snowfall over Thwaites Glacier, Antarctica. Geophysical Research Letters, 48(17), e2021GL093644.

Maclennan, M. L., Lenaerts, J. T., Shields, C. A., Hoffman, A. O., Wever, N., Thompson-Munson, M., Winters, A. C., Pettit, E. C., Scambos, T. A. & Wille, J. D. (2023). Climatology and surface impacts of atmospheric rivers on West Antarctica. The Cryosphere, 17(2), 865-881.

Masato, G., Hoskins, B. J., & Woollings, T. (2013). Winter and summer Northern Hemisphere blocking in CMIP5 models. Journal of Climate, 26(18), 7044-7059.

Mayewski PA, Frezzotti M, Bertler N, et al (2005) The International Trans-Antarctic Scientific Expedition (ITASE): an overview. Ann Glaciol 41:180–185. https://doi.org/DOI: 10.3189/172756405781813159

Mo KC, Higgins RW (1998) The Pacific–South American Modes and Tropical Convection during the Southern Hemisphere Winter. Mon Weather Rev 126:1581–1596. https://doi.org/10.1175/1520-0493(1998)126<1581:TPSAMA>2.0.CO;2

Pettersen, C., Henderson, S. A., Mattingly, K. S., Bennartz, R., & Breeden, M. L. (2022). The critical role of Euro-Atlantic blocking in promoting snowfall in central Greenland. *Journal of Geophysical Research: Atmospheres*, 127, e2021JD035776. https://doi.org/10.1029/2021JD035776

Raphael MN (2004) A zonal wave 3 index for the Southern Hemisphere. Geophys Res Lett 31:1–4. https://doi.org/10.1029/2004GL020365

Renwick JA (2005) Persistent Positive Anomalies in the Southern Hemisphere Circulation. Mon Weather Rev 133:977–988. https://doi.org/10.1175/MWR2900.1

Renwick JA, Revell MJ (1999) Blocking over the South Pacific and Rossby Wave Propagation. Mon Weather Rev 127:2233–2247. https://doi.org/10.1175/1520-0493(1999)127<2233:BOTSPA>2.0.CO;2

Schiemann, R., Athanasiadis, P., Barriopedro, D., Doblas-Reyes, F., Lohmann, K., Roberts, M. J., Sein, D. V., Roberts, C. D., Terray, L., & Vidale, P. L. (2020). Northern Hemisphere blocking simulation in current climate models: evaluating progress from the Climate Model Intercomparison Project Phase 5 to 6 and sensitivity to resolution. Weather and Climate Dynamics, 1(1), 277-292.

Sime LC, Marshall GJ, Mulvaney R, Thomas ER (2009) Interpreting temperature information from ice cores along the Antarctic Peninsula: ERA40 analysis. Geophys Res Lett 36:1–5. https://doi.org/10.1029/2009GL038982

Thomas ER, Hosking JS, Tuckwell RR, et al (2015) Twentieth century increase in snowfall in coastal West Antarctica. Geophys Res Lett 42:9387–9393. https://doi.org/https://doi.org/10.1002/2015GL065750

Turner J, Phillips T, Thamban M, et al (2019) The Dominant Role of Extreme Precipitation Events in Antarctic Snowfall Variability. Geophys Res Lett 46:3502–3511. https://doi.org/https://doi.org/10.1029/2018GL081517.

Vial, J., & Osborn, T. J. (2012). Assessment of atmosphere-ocean general circulation model simulations of winter northern hemisphere atmospheric blocking. Climate dynamics, 39, 95-112.

Winstrup M, Vallelonga P, Kjær HA, et al (2019) A 2700-year annual timescale and accumulation history for an ice core from Roosevelt Island, West Antarctica. Clim Past 15:751–779. https://doi.org/10.5194/cp-15-751-2019.

Woollings, T., Barriopedro, D., Methven, J., Son, S. W., Martius, O., Harvey, B., Sillmann, J., Lupo, A. R., & Seneviratne, S. (2018). Blocking and its response to climate change. Current climate change reports, 4, 287-300.

---

## Referee Report (RR1)

**Review Hoffman et al. 2023**

Hoffman et al. present GNSS records from the Thwaites Glacier region. They highlight the benefits of this equipment over AWS. They make composites of reanalysis parameters at the time and before extreme accumulation events. They show that blocking events with distant origins is important in blocking the westerly flow and channeling humid air masses into WA.

I suggest that the manuscript get accepted with minor revisions.

- L 196. Are these seasonal differences significant? The authors write in the response that they are significant, but it is not enough to just state that they are significant. Please, provide a p-value and which test has been used. That is, specify at what level it is significant to the main text of the article (not suppl. m.) (e.g. $p<0.05$). If it is not significant, just say that the relationship is weak, because the time series is short, and this is a bit speculative. Provide the r and p-values in the main text.

- Gaps in record. 2941 days is 8 years, and 3282 days is 9 years. Provide the length as years too, that way it is easier to see how long they are. It is a great record, but it is wrong to describe it as over 10 years of continuous data.  What are the gaps caused by storms?

- ERA5 anomalies. "Anomalies were calculated by subtracting the historical seasonal means for the observational period (2009–2022)". The whole period of reliable data 1979-2022 should be used when you calculate the anomalies, not just 2009-2002.

l. 13. …Rosby wave train…

l. 26. e.g. not i.e.

l. 44. Interannual blocking variability has been linked to marine air intrusion and accumulation variability recorded in the Roosevelt Island Ice Core, West Antarctica  AWSs were indeed used in this study but it doesn't fit here at the end of this sentence.

l. 82. Figures

l. 90. Insert a major header for the method section first "2. Methods".

l. 92 The stations' names have already been defined. No need to do it again here. Or at line 199.

l. 125 I would suggest that you don't refer to the figures here. It's enough to say something about the type of result that will come but don't go into too much detail. It is better to wait to refer to the figures in the results.

Figure 2d, is the black line in the time series from a running mean? Is it possible to make the time series less blurry? Capitalize the stations' abbreviations in the legend.

Are the supplementary figures and tables listed in the order of appearance?

Figure 3 caption. $R_h$ not RH?

l. 164 The B term could be explained earlier, eq. 6?

l. 155 Could the Metropolis-Hastings algorithm be described as machine learning? I'm not an expert in regards to the GNSS method, but your description sounds like self-learning and I see the cost function in Figure 3. If it is ML maybe write so to boost the search hits for the article.

l. 176. "Kohler Glacier site is included"

l 192. Check the header structure. The results section should have a new major header (3 Results) and not continue with 2.3.1.

l. 199. "and minimum in November…"

l. 208. Use "meters above sea level (m.a.s.l.)".

Figure 4. The figures don't appear in the order that they are presented in the text. Perhaps it would be easier if you mainly wait to introduce the figures until the results.

Figure 4. … height (grey contours)...

Fig. 4. …(cyan-green contour) with the guiding 500 hPa geopotential height (grey contours)...

Figure 4. Explain the graph to the right. Shows cumulative accumulation and the vertical dashed line indicates the timing of the atmospheric river event.

Figure 5. capitalize the letters in the station abbreviations.

Figure 6. Delete the first part here.  The seasonal accumulation cycle…

Figure 7 caption. …500-hPa geopotential height (grey contours)…atmospheric river event (cyan-green contour)

Figure 7 (D, E). Is there any significance to the fitted linear regression lines? I'm not questioning the relationship but it might not be significant for these measurements where there is a fair amount of spread in the data.

Figure 8 (A, D). Why isn't there any stippling for the blocking frequency? Make the significant test here too.

l. 231., l. 233., l.293. Remove "spring" since it is not a season that is presented results for.

l. 259. "500-hPa pressure level composites were analyzed across the Southern Hemisphere to for the significance of tropical teleconnection"?

Figure 11. "the numbering of the days indicates the days before the extreme precipitation events"

Figure 11. How come there is stippling where the anomaly is close to zero? Perhaps something spurious with the SST anomalies in the sea ice zone near Antarctica. Filter out outliers?

l. 319. Suggested change "Significant sea surface temperature composite anomalies observed in the Atlantic and western Indian Ocean preceding extreme precipitation using the GNSS-IR record are consistent with the propagation paths of eastward propagating Rossby waves modeled in the Southern Ocean by Li et al., (2015)."

P. 18-19 Disucsion about Rossby wave trains. Check if this part can be written more clearly. Split up long sentences. Can you add some explaining arrows to the plots perhaps to show the direction of the propagation?

l. 330. This is a too-sweeping general statement. Perhaps correct for the Thwaites region, the area of investigation here, but not for all Antarctica ice cores. A standard check before (or at least afterward) an ice core site is proposed would be to check for seasonal bias.

l. 335. "aquifer"?

l. 362. Delete "interrogate", "investigate" sounds better.

l. 363. "…that precedes the GNSS extreme…"

l. 372. "events"?

Supplementary material

Table 1. Column 4 name: "days (for which accumulation can be extracted and averaged)".

ERA5 anomalies. "Anomalies were calculated by subtracting the historical seasonal means for the observational period (2009–2022)". Please use the whole period of reliable data 1979-2022 when you calculate the anomalies.

Move section 3.2. Significance testing into the method section of the main text. Does the test consider autocorrelation? You give a reference for spatial autocorrelation but is regular autocorrelation in time considered?

**Comments on the author's response**

- Emanuelsson et al. first established the relationship between high-pressure anticyclones and accumulation with the RICE ice core records. Then they checked the details of the mechanism using the AWS records and ERA-interim reanalysis composites, e.g. using Z500. Then the spatial validity of the relationship was checked by checking the relationship for several West Antarctica sites: WDC, ITASE 2000- 5, and ITASE 2001-5.

---

## Author Response (AR2)

Response to major comments:

**L 196. Are these seasonal differences significant? The authors write in the response that they are significant, but it is not enough to just state that they are significant. Please, provide a p- value and which test has been used. That is, specify at what level it is significant to the main text of the article (not suppl. m.) (e.g. p<0.05). If it is not significant, just say that the relationship is weak, because the time series is short, and this is a bit speculative. Provide the r and p-values in the main text.**

The significance of this relationship was established using ERA5 by Maclennan et al., (2022). We have cited this paper and also done the P-test that the reviewers requested. For each climatology the p-value for the seasonality test is less than <.001 suggesting the seasonal cycle is very significant. We use the permutation test to assess the significance of the seasonal changes in the accumulation climatology at each site. The null hypothesis of this experiment is that the observed seasonality (or periodic pattern) is not statistically significant and could be attributed to gaussian distributed noise. We first compute the autocorrelation function for the original climatological mean and standard deviation of the accumulation time series. We then shuffle the data (randomly permuting the daily accumulation rates independently). This breaks any existing temporal structure or seasonality of the data. We then calculate the test statistic for each permutation repeating this process 10,000 times to build a distribution of the maximum autocorrelation values according to the null hypothesis.

**Gaps in record. 2941 days is 8 years, and 3282 days is 9 years. Provide the length as years too, that way it is easier to see how long they are. It is a great record, but it is wrong to describe it as over 10 years of continuous data. What are the gaps caused by storms?**

We have stated this more clearly. The total time span of the record is over 10 years, but your point about what that represents is really important. The yearly maintenance required to dig out these systems and replace the power supply is the reason there are gaps in this record, as noted above.Generally, they are not affected by storms, but by losing power when solar panels are buried or when wind generators fail (this could be storm related due to excessive winds or rime accumulation, but they are also more commonly just buried by snowfall if the site is not visited annually).

**ERA5 anomalies. "Anomalies were calculated by subtracting the historical seasonal means for the observational period (2009–2022)". The whole period of reliable data 1979-2022 should be used when you calculate the anomalies, not just 2009-2022.**

Previous work suggests that climate has changed significantly in West Antarctica over the observational period (1979-2022). This is why we focus our interpretation of renanalysis (which is done at daily resolution) to the period when we have observations.

**l. 13. …Rosby wave train…**

We appreciate the suggestion! Thank you!

**l. 26. e.g. not i.e.**

We have made the change to e.g. and been more judicious regarding proper use of these phrases throughout the manuscript.

**l. 44. Interannual blocking variability has been linked to marine air intrusion and accumulation variability recorded in the Roosevelt Island Ice Core, West Antarctica using automated weather stations. AWSs were indeed used in this study but it doesn't fit here at the end of this sentence.**

We have deleted the second part of this sentence.

**l. 82. Figures**

We have made this plural.

**l. 90. Insert a major header for the method section first "2. Methods".**

We have added the major heading. We've also changed the results section's major headings. We now have a large results section that includes subsections.

**l. 92 The stations' names have already been defined. No need to do it again here. Or at line 199.**

The sentence where these stations were first mentioned was very long. We've elected to remove their mention in this sentence and leave them listed as they are in L92. We have removed them from L199. We believe this improves readability, but are happy to make further changes as needed..

**l. 125 I would suggest that you don't refer to the figures here. It's enough to say something about the type of result that will come but don't go into too much detail. It is better to wait to refer to the figures in the results.**

Sounds good. The reference to the figures has been removed. We appreciate the suggestion.

**Figure 2d, is the black line in the time series from a running mean? Is it possible to make the time series less blurry? Capitalize the stations' abbreviations in the legend.**

The black line consists of points that represent the median filtered signal.

**Are the supplementary figures and tables listed in the order of appearance?**

They were not listed in order of appearance. We have corrected this mistake.

**Figure 3 caption. Rh not RH?**

We have changed the text in the caption to R_h.

**l. 164 The B term could be explained earlier, eq. 6?**

The B term is now explained in the section where it is first referenced (before equation 6).

**l. 155 Could the Metropolis-Hastings algorithm be described as machine learning? I'm not an expert in regards to the GNSS method, but your description sounds like self-learning and I see the cost function in Figure 3. If it is ML maybe write so to boost the search hits for the article.**

This is not a machine learning method. It's an optimization method (a more general category that is also shared by ML methods). We have more clearly referred to it now as an optimization algorithm.

**l. 176. "Kohler Glacier site is included"**

Thank you for the change.

**l 192. Check the header structure. The results section should have a new major header (3 Results) and not continue with 2.3.1.**

We have made the results section its own header and the subsections that follow have their own subsection headers.

**l. 199. "and minimum in November…"**

We've made this change and restructured this sentence.

**l. 208. Use "meters above sea level (m.a.s.l.)".**

We appreciate the suggestion. We've gone with the abbreviated "meters above sea level" as the phrase is used just twice in the article.

**Figure 4. The figures don't appear in the order that they are presented in the text. Perhaps it would be easier if you mainly wait to introduce the figures until the results.**

We have corrected this with the other changes to the figures. Thanks for the suggestion.

**Figure 4. … height (grey contours)... Fig. 4. …(cyan-green contour) with the guiding 500 hPa geopotential height (grey contours)... Figure 4. Explain the graph to the right. Shows cumulative accumulation and the vertical dashed line indicates the timing of the atmospheric river event.**

Thanks for the suggestion/catch. We have added a sentence that describes the plots with each atmospheric river we detect using the GNSS-IR time series.

**Figure 5. capitalize the letters in the station abbreviations.**

Letters have been capitalized. Thanks for the catch!

**Figure 6. Delete the first part here. Interannual accumulation and t The seasonal accumulation Cycle…**

We have deleted the first part of this sentence.

**Figure 7 caption. …500-hPa geopotential height (grey contours)…atmospheric river event (cyan-green contour)**

We have made the suggested change. Thank you!

**Figure 7 (D, E). Is there any significance to the fitted linear regression lines? I'm not questioning the relationship but it might not be significant for these measurements where there is a fair amount of spread in the data.**

The fit is significant and is now accurately cited in the text.

**Figure 8 (A, D). Why isn't there any stippling for the blocking frequency? Make the significant test here too.**

Blocking frequency requires a bootstrapping approach that was beyond the scope of this study to calculate as blocking itself is a binary field.

**l. 231., l. 233., l.293. Remove "spring" since it is not a season that is presented results for.**

We do have springtime blocking included in the supplement; this is now referenced more carefully here.

**l. 259. "500-hPa pressure level composites were analyzed across the Southern Hemisphere to for the significance of tropical teleconnection"?**

We've changed the structure of this sentence to be more accurate and briefer following the reviewer's suggestion.

**Figure 11. "the numbering of the days indicates the days before the extreme precipitation events"**

We have changed this sentence to respond to the awkward phrasing of the indexed days.

**Figure 11. How come there is stippling where the anomaly is close to zero? Perhaps something spurious with the SST anomalies in the sea ice zone near Antarctica. Filter out outliers?**

Anomalies close to zero can still be significant if the relationship is weak. We have elected to keep these,but can remove them if the editor wishes to have a threshold tolerance for the relationship.

**l. 319. Suggested change "Significant sea surface temperature composite anomalies observed in the Atlantic and western Indian Ocean preceding extreme precipitation using the GNSS-IR record are consistent with the propagation paths of eastward propagating Rossby waves modeled in the Southern Ocean by Li et al., (2015)."**

We agree with this change and have made it in the text. Thank you

**P. 18-19 Disucsion about Rossby wave trains. Check if this part can be written more clearly. Split up long sentences. Can you add some explaining arrows to the plots perhaps to show the direction of the propagation?**

We have edited this section and the plots according to the reviewer suggestions.

**l. 330. This is a too-sweeping general statement. Perhaps correct for the Thwaites region, the area of investigation here, but not for all Antarctica ice cores. A standard check before (or at least afterward) an ice core site is proposed would be to check for seasonal bias.**

Some of these Thwaites sites are closer to WAIS divide then they are to the grounding zone. We agree that these can't be generalized to all Antarctic cores. We do think this applies to WAIS cores, and we also think the GNSS-IR method is a really great way to

**l. 335. "Aquifer"?**

Yes, thanks for the suggestion. We have made the spelling change.

**l. 362. Delete "interrogate", "investigate" sounds better.**

Thanks for the suggestion. We agree.

**l. 363. "…that precedes the GNSS extreme…"**

We've rewritten the sentence combining extreme precipitation with vapor transport as processes explored using the GNSS-IR accumulation time series.

**l. 372. "events"?**

Thanks for the catch.

**Supplementary material**
**Table 1. Column 4 name: "days (for which accumulation can be extracted and averaged)". ERA5 anomalies. "Anomalies were calculated by subtracting the historical seasonal means for the observational period (2009–2022)". Please use the whole period of reliable data 1979-2022 when you calculate the anomalies.**

We do not use the whole period as there are trends in these data. The mean from the modern period seems more consistent with the anomaly associated with the events that we catalog and is consistent with other observational reanalysis studies.

**Move section 3.2. Significance testing into the method section of the main text. Does the test consider autocorrelation? You give a reference for spatial autocorrelation but is regular autocorrelation in time considered?**

This is an autocorrelation in time at spatial points closest to each site.

**Emanuelsson et al. first established the relationship between high-pressure anticyclones and accumulation with the RICE ice core records. Then they checked the details of the mechanism using the AWS records and ERA-interim reanalysis composites, e.g. using Z500. Then the spatial validity of the relationship was checked by checking the relationship for several West Antarctica sites: WDC, ITASE 2000- 5, and ITASE 2001-5.**

We have highlighted this result in the text. The WDC and ITASE 2000-5 and ITASE 2001-5 cores sites are deeper into the interior than these sites that are 100 km closer to the coast. Our records also record snowfall directly and so issues associated with firn densification do not need to be considered.